# Demographic Analysis of Shortfin Mako Shark (*Isurus oxyrinchus*) in the South Pacific Ocean

**DOI:** 10.3390/ani12223229

**Published:** 2022-11-21

**Authors:** Hoang Huy Huynh, Chun-Yi Hung, Wen-Pei Tsai

**Affiliations:** 1Institute of Aquatic Science and Technology, National Kaohsiung University of Science and Technology, Kaohsiung 81157, Taiwan; 2Department of Fisheries Production and Management, National Kaohsiung University of Science and Technology, Kaohsiung 81157, Taiwan; 3Division of Fisheries Ecology and Aquatic Resources, Research Institute for Aquaculture No. 2, Ho Chi Minh 710000, Vietnam

**Keywords:** demographic analysis, data-limited, Monte Carlo simulation, shortfin mako shark, two-sex stage-based matrix model, uncertainties

## Abstract

**Simple Summary:**

Statistical of baseline and species-specific data are largely unavailable for several elasmobranchs (e.g., sharks, skates, and rays). Populations of many shark species such as shortfin mako sharks (*Isurus oxyrinchus*) have declined, mainly because of overexploitation and fishing pressure-related vulnerability. Mako shark conservation and management have become major concerns. The available catch, fishing effort, and biological composition data of mako sharks, particularly in the South Pacific Ocean, are insufficient, making stock assessments difficult. We applied demographic analysis to increase the current understanding of the mako shark stock status despite insufficient fishery data. Our results revealed that mako shark populations would only be able to sustain a lower fishing mortality (20% of natural mortality) in the study area. The findings strongly suggest that more conservative management measures should be implemented to ensure sustainable utilization of the mako shark stock. The current models may be applicable to other shark species and taxa with limited data availability for preserving their ecological balance and establishing management and conservation measures for them.

**Abstract:**

The shortfin mako shark (*Isurus oxyrinchus*) demonstrates low productivity and is thus relatively sensitive to fishing. Natural mortality (*M*) and fishing mortality (*F*) data are critical to determine their population dynamics. However, catch and fishing effort data are unavailable for this species in the South Pacific Ocean, making stock assessments difficult. Demographic quantitative methods aid in analyzing species with limited data availability. We used a two-sex stage-structured matrix population model to examine the demographic stock status of mako sharks. However, data-limited models to determine fishery management strategies have limitations. We performed Monte Carlo simulations to evaluate the effects of uncertainty on the estimated mako shark population growth rate. Under unfished conditions, the simulations demonstrated that the mako sharks showed a higher finite population growth rate in the 2-year reproductive cycle compared to the 3-year reproductive cycle. Protecting immature mako sharks led to a higher population growth rate than protecting mature mako sharks. According to the sex-specific data, protecting immature male and female sharks led to a higher population growth rate than protecting mature male and female sharks. In conclusion, sex-specific management measures can facilitate the sustainable mako shark conservation and management.

## 1. Introduction

Globally, the scientific evaluation of many overexploited marine species, particularly sharks, is lacking, which has prevented their sustainable exploitation and effective conservation [1,2,3]. In recent years, various regional fisheries management organizations such as the International Commission for the Conservation of Atlantic Tunas (ICCAT), the Indian Ocean Tuna Commission (IOTC), and the Western and Central Pacific Fisheries Commission (WCPFC) have conducted shark stock assessments and have consequently placed restrictions on the commercial retention of various shark species including the shortfin mako shark (hereafter, referred to as “mako shark”). However, the Inter-American Tropical Tuna Commission (IATTC) is yet to report its shark stock assessment results. The management measures implemented by the aforementioned organizations indicate the urgent need for an improved understanding of shark stock status.

The mako shark (*Isurus oxyrinchus* Rafinesque, 1810, from the family Lamnidae) is fast-swimming, and is a vital component of the pelagic shark community and is widely distributed in tropical and temperate waters with a temperature of >16 °C [4,5]. In 2019, it was classified as Endangered (EN) on the International Union for Conservation of Nature (IUCN) Red List [6] and was included in the Convention on International Trade in Endangered Species of Wild Fauna and Flora (CITES) Appendix II [7]. This species is known for its long life, low fecundity [8,9], late maturity [10,11], and extended reproductive cycle (2–3 years) [9,10,12]. Consequently, it is extremely susceptible to overexploitation. Given its high susceptibility and low productivity, the mako shark has been identified as one of the most vulnerable species in the Atlantic, Indian, and Pacific Oceans, as per an ecological risk assessment [13,14,15].

Shark conservation status demonstrates considerable variations across regions, and shark population trends are unpredictable. Several recent studies have focused on mako shark stock status in various regions worldwide including in the Atlantic Ocean [16], northwestern Pacific Ocean [17,18,19,20], and Indian Ocean [21,22]. However, apart from two recent preliminary mako shark stock assessment reports in the Southwest Pacific [23,24], its population status remains poorly known, and in terms of biological information, only three investigations have been conducted to date [25,26,27]. Obtaining the long-term catch and effort data of fish is difficult, and the use of biomass dynamics models (e.g., Bayesian biomass dynamics model, virtual population analysis, or surplus production model), which have been used to assess the bigeye thresher shark (*Alopias superciliosus*) [28] and mako shark [17], can be complicated. Similarly, age-structured assessment models (e.g., stock synthesis) such as that used for bigeye tuna (*Thunnus obesus*) [29] generally require steepness values of stock–recruitment relationships. The traditional Leslie matrix method (based on an age and stage–based model), along with deterministic and stochastic approaches, have been used to assess the stock status of pelagic thresher sharks (*Alopias pelagicus*) [30,31], the silky shark (*Carcharhinus falciformis*) [32], the mako shark [19,20], and the blue shark (*Prionace glauca*) [33]. Data-limited management methods are increasingly being used to reflect the state of fishery areas [34,35,36], and demographic models may be used to provide valuable insights into the management recommendations for these species.

In fisheries, demographic models have several advantages over conventional stock assessment models. Traditional stock assessment approaches, surplus production models, or age-structured population models require a large amount of data (e.g., catch, effort, and abundance indices). In contrast to conventional models, demographic models require only data-limited life history parameters (i.e., survival rate, age at maturity, longevity, litter size, and number of embryos) [37,38]. Many elasmobranch populations in the Pacific and Atlantic Oceans have been studied using these models [16,19,30,39,40,41,42,43]. In addition, these models are more applicable for long-living, slow-growing shark species [42] such as the mako shark. They can compensate for data imbalance and hence serve as a valuable assessment tool until sufficient fisheries data for traditional stock assessments are available or until alternative stock assessment methods are developed [37,38]. Moreover, marine fish growth patterns are sex-specific and dimorphic, as in the mako shark [9,17,18,19,44,45], boarfish (*Capros aper*) [46], electric knifefish (*Gymnorhamphichthys rondoni*) [47], and European sea bass (*Dicentrarchus labrax* L.) [48]. Two-sex models are applied when species (e.g., the mako shark) exhibit considerable sexual dimorphism [45,49,50,51].

Acquiring species-specific biological information for stock assessment is critical to ensure that shark populations are exploited sustainably. For this purpose, by using Monte Carlo simulations, we constructed a stochastic framework accounting for uncertainty in crucial parameters; this is the standard methodology currently used to assess shark populations [19,38,45,52,53]. The objectives of the current study were to **(i)** develop and use a two-sex stage-based matrix model with Monte Carlo simulations to assess the uncertainty in the life history parameters of mako sharks; **(ii)** estimate their finite rate of population growth (*λ*) and the intrinsic rate of population increase (*r*); **(iii)** generate various scenarios to assess the influence of their fishing mortality (*F*) using many approaches to assess natural mortality (*M*); and **(iv)** compare their demography in other oceans. In the current study, we, for the first time, provide information applicable to mako shark population management and conservation in the South Pacific Ocean by using a demographic methodology; our methods may be broadly applicable to other data-limited shark species as well as to other taxa.

## 2. Materials and Methods

### 2.1. Study Area and Biological Parameters

This study was conducted in waters in the South Pacific Ocean, extending from 0°0′0″ to 60°0′0″ S and from 130°0′0″ E to 70°0′0″ W (Figure 1), and we assumed the existence of a unit stock for mako sharks to simplify the development of the matrix model. The values of the biological parameters for mako sharks in the South Pacific Ocean have been estimated by Chan [25], Bishop et al. [26], and Cerna and Licandeo [27]. However, Cerna and Licandeo [27] provided parameters with a wider range and estimates with higher accuracy. Therefore, these parameters were used as the foundation for the current demographic study (Table 1).

### 2.2. Mako Shark Life History

The reproductive biology of mako sharks in the study area was provided by Bishop et al. [26]. The life phases and duration of mako sharks were established by Tribuzio and Kruse [42], who examined their life history; the developmental stages of mako sharks include the neonate, juvenile, subadult, and adult stages. Mature female mako sharks can be either pregnant (adult pregnant) or not pregnant (adult resting); they typically alternate between the two states. A female in the resting stage must return to the pregnant stage after 1 year, and a pregnant female may return to the pregnancy stage or move to the resting stage [54]. Gestation lasts 15–18 months [55,56]; in other words, pregnancy in a female adult mako shark lasts for approximately 2 years.

In the present study, the life history of female mako sharks was divided into six stages: neonate (0–1 year), juvenile (1–17 years), subadult (17–21 years), pregnant adult (1 year), parturient adult (1 year), and resting adult (1 year); only in case of a 3-year reproductive cycle. Moreover, the life history of male mako sharks was divided into four stages: neonate (0–1 year), juvenile (1–6 years), subadult (6–10 years), and adults (>10 years; Figure 2 and Table 2). According to the recommendations of Duffy and Francis [57], female sharks have a 3-year reproductive cycle with a 2-year gestation period and a 1-year resting period; however, some of them may have a 2-year reproductive cycle [10]. Consequently, we included both the 2- and 3-year reproductive cycles in our analysis.

### 2.3. Model Development

We used a stage-structured model as a component of an ungenerous technique to estimate allowed injuries because it has been applied to the majority of threatened species with limited age-specific data. Management strategies are more likely to be related to life cycle stages than to age classes. The demographics of the mako shark in the South Pacific Ocean were therefore examined using a stage-structured Leslie matrix population projection model [58]. This is the fundamental equation to estimate the population’s stage structure at any time *t*:(1)$$N_{t+1}=A_{t}N_{t}$$
where *N_t_* is the vector of the number of sharks in each stage class at time *t*, and *A_t_* is the life-history projection matrix comprising survival and fecundity for each stage at time *t* [37,59].

Our model, nevertheless, assumed that harvesting occurred before natural mortality (*M*) and reproduction. As the highly migratory mako shark is found over a wide range of area throughout the South Pacific Ocean, any regulatory theory based on competition for resources in a space-limited environment is more suitable than a hypothesis. Thus, the dynamics of density dependence on parameters such as survival, fecundity, and growth were ignored in this study. A stage-structured model based on maturity or breeding conditions involves the concept of knife edge (i.e., step-like) changes from one stage to another. Hence, knife-edge maturity was applied in this model.

Stage-based projection models are typically used based on their ability to analytically solve *A_t_* to derive significant demographic variables and determine their sensitivity to the parameter estimations. Here, we established two-sex models in which the abundance of both male and female sharks influences fecundity [19,45,49] in the 2-year reproductive cycle and the 3-year reproductive cycle with a 1-year resting period; these are shown in Equations (2) and (3), respectively:(2)$$A_{t}=\left[\begin{array}{cccccccc} 0 & 0 & 0 & f_{4}^{mal} & 0 & 0 & 0 & G_{8}^{fem}\times f_{8}^{fem}\\ \rho G_{1} & p_{2}^{mal} & 0 & 0 & 0 & 0 & 0 & 0\\ 0 & G_{2}^{mal} & p_{3}^{mal} & 0 & 0 & 0 & 0 & 0\\ 0 & 0 & G_{3}^{mal} & p_{4}^{mal} & 0 & 0 & 0 & 0\\ (1-\rho )G_{1} & 0 & 0 & 0 & p_{5}^{fem} & 0 & 0 & 0\\ 0 & 0 & 0 & 0 & G_{5}^{fem} & p_{6}^{fem} & 0 & 0\\ 0 & 0 & 0 & 0 & 0 & G_{6}^{fem} & 0 & G_{8}^{fem}\\ 0 & 0 & 0 & 0 & 0 & 0 & G_{7}^{fem} & 0 \end{array}\right]$$
(3)$$A_{t}=\left[\begin{array}{ccccccccc} 0 & 0 & 0 & f_{4}^{mal} & 0 & 0 & 0 & G_{8}^{fem}\times f_{8}^{fem} & 0\\ \rho G_{1} & p_{2}^{mal} & 0 & 0 & 0 & 0 & 0 & 0 & 0\\ 0 & G_{2}^{mal} & p_{3}^{mal} & 0 & 0 & 0 & 0 & 0 & 0\\ 0 & 0 & G_{3}^{mal} & p_{4}^{mal} & 0 & 0 & 0 & 0 & 0\\ (1-\rho )G_{1} & 0 & 0 & 0 & p_{5}^{fem} & 0 & 0 & 0 & 0\\ 0 & 0 & 0 & 0 & G_{5}^{fem} & p_{6}^{fem} & 0 & 0 & 0\\ 0 & 0 & 0 & 0 & 0 & G_{6}^{fem} & 0 & 0 & G_{9}^{fem}\\ 0 & 0 & 0 & 0 & 0 & 0 & G_{7}^{fem} & 0 & 0\\ 0 & 0 & 0 & 0 & 0 & 0 & 0 & G_{8}^{fem} & 0 \end{array}\right]$$

Here, the male and female parameters of matrix *A_t_* are indicated using the subscripts *mal* and *fem*, respectively, and ρ denotes the sex ratio at birth (set at 0.5 to achieve an equal offspring sex ratio) [10]. In the post-breeding survey, the fertility coefficient *f*_i_ measures the stage-specific per-capita fecundity, which can be calculated as follows: *f_i_* = *fi* × *σ_i_* [59]. Caswell’s [59] modified harmonized mean birth function was used to analyze the influence of the sex ratio on the mating success, as shown in Equations (4) and (5), respectively:(4)$$f_{4}^{mal}(n)=\frac{kn_{f}}{n_{f}+n_{m}}$$
(5)$$f_{8}^{fem}=\frac{kn_{m}}{n_{f}+n_{m}}$$
where *k* denotes the litter size and *n_m_* and *n_f_* denote the reproductive densities of male and female sharks, respectively. Considering the lack of evidence on the mating systems of mako sharks, monogamous mating and an equal litter size for both sexes were assumed in Equations (4) and (5)—independent of size or even other stage characteristics.

*G_i,s_* is the product of the probability of an individual in stage *i* surviving (*σ_i,s_*) and the probability of shifting to another stage (*γ_i,s_*) for each sex: $G_{i,s}=\sigma_{i,s}\times \gamma_{i,s}$; moreover, *p_i,s_* is the probability of an individual surviving and remaining in its current stage: $p_{i,s}=\sigma_{i,s}\times (1-\gamma_{i,s})$ [54,60]. For a single stage, *σ_i,s_* is denoted as $\sigma_{i,s}=e^{-(M_{i,s}+F_{i,s})}$, here, *F_i,s_* is the *F* at stage *i* and is set as the average over-age of age-specific *F* (*F_a,s_*), and *M_i,s_* is the *M* at stage *i*, which is set as the average over-age of age-specific natural mortality. *γ_i,s_* is calculated as follows:(6)$$\gamma_{i,s}=\frac{{\left(\sigma_{i,s}/\lambda_{init}\right)}^{T_{i,s}}-{\left(\sigma_{i,s}/\lambda_{init}\right)}^{T_{i,s}-1}}{{\left(\sigma_{i,s}/\lambda_{init}\right)}^{T_{i,s}}-1}=\frac{{\left(\sigma_{i,s}\right)}^{T_{i,s}}-{\left(\sigma_{i,s}\right)}^{T_{i,s}-1}}{{\left(\sigma_{i,s}\right)}^{T_{i,s}}-1}$$
where *λ_init_* is the initial population growth rate (=1), and *T_i,s_* is the duration of each sex and stage. An iterative approach was used to estimate the matrix parameters using Equation (6). The finite population growth rate (*λ*) can be obtained by solving the following Equation:(7)$$\left|A-\lambda I=0\right|$$
where *I* is the identify matrix [59]. *λ* for each sex can be calculated by separating *A_t_* into the matrices for male and female sharks, as illustrated in Equations (2) and (3). Moreover, the intrinsic rate of population growth (*r*) may be related to *λ* obtained through the matrix projection [37]:(8)$$\lambda =e^{r}$$

The net productive rate *R_0_* is the mean number of pups produced by which a newborn individual is replaced by the end of its life. Generation time (*T*) [61] is defined as the time required for the population to increase by a factor of *R*_0_ and can be calculated as follows:(9)$$T=\frac{\ln R_{0}}{\ln\lambda }=\frac{\ln R_{0}}{r}$$

Here, when the finite rate of population growth *λ* = 1, the population is stationary; when *λ* > 1, overpopulation has occurred (this is when harvesting may be considered an alternative to maintaining a stable population); and when *λ* < 1, the population is declining.

The relative proportion of individuals in each stage remains consistent throughout time after obtaining the stable stage distribution (SSD) vector *w*, which is a right eigenvector and one of the two demographic traits that can be derived from *A_t_*. The reproductive value (RV) vector *v* is a left eigenvector, which denotes the proportion of offspring that individuals have yet to give birth at a given age [62]. RV depends on the number of individuals surviving to maturity and reproduction. Thus, individuals in the earliest stages should have the lowest RVs because they must survive and reach maturity before reproducing. An eigen analysis can be used to calculate the *w* and *v* of *A_t_*. For any {n × n} matrix, up to n scalar values (*λ*_1…n_) and n-associated right and left vectors so that:(10)$$Aw=\lambda w,\;vA^{T}=\lambda v$$
where *A^T^* is the transpose of *A_t_*; and *w* and *v* are the right and left eigenvectors of *A_t_,* respectively.

### 2.4. Mortality Estimation

Mortality is a crucial parameter for understanding the dynamics of any population, and it should be estimated for mako sharks. Calculation of the natural mortality (*M*) of shark species is difficult [63,64]. The magnitude of *M* is closely related to population productivity, environmental parameters, exploitation rates, management variables (e.g., maximum sustainable yield), and biological reference points [63,65] and it is frequently the main source of uncertainty in population dynamics modeling and stock assessment.

In this study, mako shark data and methods for the direct estimation of their *M* were unavailable. Therefore, we estimated *M* by using indirect methods. Sex-specific *M* (*M_s_*) was assumed to remain constant in the subsequent deterministic analyses, which were based on the median obtained of the 14 methods listed in Table 3.

### 2.5. Demographic Methods Accounting for Uncertainty

As with many shark species, estimating the rates of vital parameters is difficult due to limited data and vulnerability to uncertainty, so accounting for the effects of uncertainty by integrating stochasticity in demographic models is essential [74]. In the current study, Monte Carlo simulations were used [52,75]. This study accounted for uncertainty by using a simulation methodology similar to that used by [52], where statistical distribution functions based on published data were defined for each life history and biological parameter. Because detailed historical catch and effort data for mako sharks in the South Pacific Ocean are rare, each significant parameter was calculated with (*Z* = % of considered *M*) and without (*Z = M + F*) for the 2- and 3-year reproductive cycles; here, *Z* is the total mortality rate of a population.

Fecundity is a crucial component in demographic analysis. In several studies, the majority of observations lies between 8 and 18, and the mean litter size for the studied species was determined to be 12.5 [8,10,76,77,78]; moreover, the triangular distribution including the lower and upper limits and the assignment of a most likely value between these limits was assumed.

In the case of *M_s_* uncertainty, the mean *M_s_* values and their standard deviation (SD) were obtained in each scenario using the 14 indirect techniques for each sex and stage indicated in Table 3; these scenarios were used to generate a lognormal distribution. A lognormal error structure for *M_s_* calculation ensures that the converted estimates are positive and between 0 and 1.

We assumed the lower and upper bounds of the uniform probability distribution approach in the case of age at maturity uncertainty to be 7–10 in male sharks and 19–21 in female sharks.

In the presence of longevity uncertainty, theoretical longevity estimates were based on Equations (11) and (12) for male sharks [79,80] and females [80,81], respectively:(11)$$T_{\max}=7\ln\left(\frac{2}{k}\right)\left[79\right],\;T_{\max}=t_{0}-\frac{\ln(0.05)}{k}$$
(12)$$T_{\max}=\frac{1}{k}\ln\left[\frac{\left(L_{\infty }-L_{0}\right)}{L_{\infty }(1-x)}\right],\;T_{\max}=\frac{1}{k}\ln\left[\frac{\ln(L_{0}/L_{\infty })}{\ln(x)}\right]\left[81\right]$$
where *k*, *L_∞_* and *L*_0_ are the growth parameters of the VBGE [27]; and *x* is the proportion of *L_∞_* reached at *T*_max_, which is commonly assumed to be 0.95 [80,82]; the estimator with an estimated *T*_max_ using *x = 0.95* is hereafter referred to as *L_∞_ 95*. These theoretical estimated values were used to define the upper and lower bounds of a discrete uniform distribution of longevity. The uncertainties involved in fecundity, *M_s_*, age at maturity, and longevity were evaluated using corresponding assumed distributions and are displayed in Table 4.

### 2.6. Size Limit Measures

Estimating the current fishing levels is difficult mainly because accurate harvest data are lacking; this hinders quantitative stock assessments for mako sharks in the study area. As the mako shark demonstrates considerable sex and size segregation on the area scale [9,83], along with segregation between juveniles and other stages [84], setting size limits for mako sharks may be simple. Consequently, we conducted simulations to analyze the existing status and to evaluate various management strategies by generating different *F* values based on a range of methodologies in the theoretical estimations of *M_s_*.

To investigate stock status, 11 scenarios of harvest strategy were established in this study. For both the 2- and 3-year reproductive cycle models, five scenarios for harvest strategies and six scenarios for protective measures were as follows:Scenario 1 (S1): Under natural conditions, *F* was set to 0 for all ages;Scenario 2 (S2): *F* was set to 80% of *M_s_* by stage;Scenario 3 (S3): *F* was set to 60% of *M_s_* by stage;Scenario 4 (S4): *F* was set to 40% of *M_s_* by stage;Scenarios 5 (S5): *F* was set to 20% of *M_s_* by stage;Scenario 6 (S6): With the protection of immature sharks, *F* was set to 80% of *M_s_* by stage; *F* values of neonates, juveniles, and subadults of both sexes were set to 0;Scenario 7 (S7): With the protection of mature sharks, *F* was set to 80% of *M_s_* by stage, and *F* values of adults of both sexes were set to 0;Scenario 8 (S8): With the protection of immature male sharks, *F* was set to 80% of *M_s_* by stage, *F* values of male neonates, juveniles, and subadults were set to 0;Scenarios 9 (S9): With the protection of mature male sharks, *F* was set to 80% of *M_s_* by stage, *F* values of male adults was set to 0;Scenario 10 (S10): With the protection of immature female sharks, *F* was set to 80% of *M_s_* by stage; *F* values of female neonates, juveniles, and subadults were set to 0;Scenario 11 (S11): With the protection of mature female sharks, *F* was set to 80% of *M_s_* by stage; *F* value of female adults was set to 0.

The parameter values were calculated as the means of 10,000 replicates, with 95% confidence intervals (CIs) being the 2.5th and 97.5th percentiles. *λ* and *r* were obtained using Monte Carlo simulations accounting for parameter uncertainty for each scenario. All the aforementioned demographic and simulation analyses were performed using PopTools [85], and programmed using R [86].

## 3. Results

### 3.1. M_s_ Estimation

In all circumstances, male sharks had higher estimated *M_s_* values than female sharks (Figure 3)—with the median *M_s_* being 0.156 and 0.126 year^−1^, respectively. *M_s_* values obtained using the 14 estimators in Table 3 for male and female sharks are illustrated in Appendix A. As shown in Figure 3, compared with other estimators, M3 and M14 demonstrated significant differences in male sharks, and M3 and M9 exhibited significant differences in the female sharks.

### 3.2. Demographic Analysis for the 2- and 3-Year Reproductive Cycle Models

Given the uncertainty in the mortality estimates and life history parameter availability, a range of scenarios, all containing stochasticity (noted through the Monte Carlo simulation), were examined. According to the most optimistic input parameters, the deterministic model of unfished conditions indicated an increasing population with mean and 95% CI for our 2- and 3-year reproductive cycle models, with sex-specificity evident in all scenarios (Appendix A).

Under unfished conditions, the stage-structured model based 2- and 3-year reproductive cycle models exhibited equivalent population growth with *λ* > 1 (1.098 and 1.063 year^−1^, respectively); however, the net reproduction rates, *R_0_*, differed slightly (3.630 and 2.472, respectively) because the duration spent in the adult stages differed (Appendix A). The *λ* estimates for the 2-year reproductive cycle were higher than those for the 3-year reproductive cycle. In addition, without *F*, the mean mako shark generation time, *T*, (95% CI) was anticipated to be 13.468 (10.874–16.518) and 14.260 (11.491–17.584) years in the 2- and 3-year reproductive cycle models, respectively. When *F* was incorporated into the models (Scenarios 2–5), all the 2- and 3-year reproductive cycle models tended to provide a reduced value of *λ*. Our results further indicate that under Scenario 5 (0.2 *M_s_*; *λ* of >1) and the given harvest conditions (*F* = 20% of *M*), a population of mako sharks would be stable (Appendix A).

In summary, the aforementioned findings confirmed that the 3-year reproductive cycle model estimated a lower population growth rate compared with the 2-year reproductive cycle, particularly in female sharks. Moreover, the estimated elements of *A_t_* for both reproductive cycle models under unfished conditions are presented in Appendix A.

### 3.3. SSD and RV

The stable stage distribution (SSD) and the reproductive value (RV) of our 2- and 3-year reproductive cycle models demonstrated some similarities and differences. In general, the SSDs were extremely similar across the five harvest strategy scenarios, and the RVs varied across the scenarios. Although the SSDs shifted mainly to juveniles in both male and female sharks in both reproductive cycle models, the general pattern of stage distributions remained unchanged (Figure 4).

For all scenarios investigated, RVs tended to increase in the 2-year reproductive cycle model, peaking in adult male and adult parturient female sharks (Figure 5a). In our 3-year reproductive cycle model, RVs also tended to increase, peaking in adult male and adult parturient female sharks, but they decreased in adult resting female sharks (Figure 5b).

Moreover, the neonates in the stage-based model were expressed as a percentage of the population, which was approximately 20% (Figure 4). Although the RV for all stages probably exhibited a nearly bell-shaped curve indicating changes between stages, the RVs for neonates, juveniles, and subadults were low, and almost all RV values were observed for adult male sharks, adult pregnant females, and adult parturient female sharks for the 2- and 3-year reproductive cycle models, respectively, and adult resting female stages (only in a 3-year reproductive cycle) (Figure 5).

### 3.4. Size Limits Measures

In general, the *λ* values for mako sharks fluctuated considerably across the different management measure scenarios evaluated in the demographic models. Compared with the 3-year reproductive cycle model, the 2-year reproductive cycle model estimated higher management measurements including the population growth rates (Figure 6 and Figure 7). Furthermore, because of the parameter uncertainty, the *λ* values of both sexes, males and females, exhibited significant variation in most of the scenarios.

The simulation results demonstrated that the stock status under 20% fishing pressure (Scenario 5) led to a stable population growth rate in our 2- and 3-year reproductive cycle models (Figure 6a,b). However, *λ* was the lowest in female sharks (0.991 year^−1^ in Scenario 5 in the 3-year reproductive cycle model), which was lower than the threshold *λ* of 1. Notably, under the *F* put into these models (Scenarios 2 and 3) in both the reproductive cycle models, all *λ* values were <1, particularly for female sharks in Scenarios 2–4.

In both the reproductive cycle models, the population increases were noted when immature mako sharks were protected (Scenario 6; Figure 7a,b). Both the reproductive cycles predicted a higher population growth rate when immature male and female sharks were protected than when mature male and female sharks were protected (Figure 7). However, under these measures, the female stocks demonstrated a significant decline when immature male sharks were protected (Scenario 8; Figure 7a,b), particularly when *λ* was relatively lower than 1. Even when mature male sharks were protected in Scenario 9 (Figure 7), the population growth rate of the female stock was possibly <1. Variance in the aforementioned scenarios was due to the influence of uncertainty in age at maturity and the longevity of male and female sharks in the South Pacific Ocean.

## 4. Discussion

### 4.1. M_s_

In this study, despite limited data availability, we estimated the *M_s_* of mako sharks in the South Pacific Ocean by using indirect (empirical) approaches. Several indirect approaches have been proposed for estimating the *M* of fish including some proposed for species with limited data availability such as that reported by Kenchington [64] as well as some newly developed methods. Estimating *M*—an essential parameter in population dynamics—is difficult [87]. Therefore, we employed 14 previously reported empirical estimators along with a detailed assessment of uncertainty with the related parameters. Although Brodziak et al. [63] and Hoenig et al. [87] recommended comparing indirect estimates of *M* with the field-derived measurements of *M*, direct estimates of *M* are unavailable due to the lack of mako shark capture and effort data. In the current study, in the South Pacific Ocean, male sharks had a higher *M_s_* than female sharks (0.106–0.246 vs. 0.069–0.166 year^−1^), similar to the data obtained at other locations: Takeuchi et al. [16] reported that based on the average mortality estimation methodology, the *M_s_* of mako sharks in the Atlantic Ocean was 0.122–0.168; moreover, the *M_s_* of male and female mako sharks in the Northwest Pacific Ocean was estimated to be 0.093–0.200 and 0.077–0.242, respectively, according to Tsai et al. [18] and 0.119–0.141 and 0.091–0.124 according to Tsai et al. [19].

Substantial variation in the *M* values is related to the indirect approach used; these approaches differ in terms of the number of parameters used, their uncertainty, and their theoretical foundation. For instance, two life history parameters were considered in M14 (*k* and *L_∞_*) [68] and M8 (age at maturity and *t_0_*) [41]. M12 [72] and M13 [69] considered only the somatic growth rate *k*; finally, M1 [66] and M2 [67] considered only the longevity parameter. Empirical methods are often perceived to be less reliable than direct methods such as mark-recapture [88,89] and telemetry [90,91]; moreover, the measurement of life history parameters can be erroneous [69,92].

### 4.2. Demographic Analysis for Mako Shark Stock Assessment

Demographic analysis of long-living, highly migratory species such as mako sharks has limitations. Nevertheless, this study is the first to assess mako shark stock in the South Pacific Ocean through demographic analysis and to forecast how mako shark populations respond to alternative management strategies. Demographic analysis of shark populations can facilitate the effective management of their stocks and the assessment of their population dynamics. This may be used to estimate the most fundamental statistical parameters related to shark stock fluctuation and management: *r* and *λ*. Moreover, we constructed a two-sex stage-structured matrix population model accurately reflecting mako shark life cycles to examine the influence of biological parameters on the population growth rate.

According to our demographic estimates obtained using the 2- and 3-year reproductive cycle models, mako sharks are one of the least productive shark species. In the Ecological Risk Assessment (ERA) ranking for longline gear, mako sharks have been allocated the highest vulnerability level (No. 1) [93]. In the models without *F*, the mean population growth rates in this study (*λ* = 1.098 and 1.063 year^−1^ for the 2- and 3-year reproductive cycle models, respectively) were moderately higher than those derived from previous demographic analyses [20,94,95]; moreover, they differed considerably from those reported by Cortés [52] (Table 5). The reasons underlying these differences are diverse, with the most typical being the use of different models (single-sex vs. two-sex, age-structured vs. stage-structured, Bayesian surplus production model) and life history parameters (e.g., *M_s_* and population growth rate) in the aforementioned studies; moreover, the findings for mako sharks in the South Pacific Ocean may not be applicable to those in other oceans, since the shark populations in the different oceans have varying traits.

Conventional demographic analysis assumes that male sharks do not control population growth; most studies have used the data of only female sharks. Two-sex models are, however, required to analyze the ecological and evolutionary implications of sexual dimorphism [96]; this is because single-sex models provide unsatisfactory results if demographic parameters such as size at age differ across sexes [49,50,97]. Consequently, male and female dynamics may differ, necessitating the incorporation of the data of both sexes in population models to understand population dynamics. The current findings demonstrate that when sex is considered in the model, the population growth rates are higher compared to those reported previously [19,21,45]. However, Yokoi et al. [53] reported values greater than this study’s demographic parameters because of the age at maturity and longevity differences across the studied oceans. Therefore, this study confirmed that age at maturity and longevity were significantly and negatively correlated with the population growth rate of sharks [30,52,79].

The demographic method can be used to establish the prior distribution of important parameters in stock assessment (e.g., *r*), especially for parameters required in Bayesian stock assessment [98]. These population growth rate values can be used as the prior distribution or input variables for the Bayesian surplus production model (BSPM) under the stock assessment framework and ERA productivity index. For instance, Tsai et al. [99] estimated bigeye thresher shark population development by using a Bayesian population model; moreover, the vulnerability of pelagic sharks and mako sharks has also been assessed using ERA [100,101]. The present findings may aid in performing similar assessments. The aforementioned BSPM, for example, has been used to calculate the stock sizes of mako sharks in the South and North Atlantic Oceans. Thus, the high population growth rate estimated here can improve the future stock estimates of mako sharks in the South Pacific Ocean, at least until sufficient fishery data become available.

Variations in biology, morphology, behavior, and reproductive functions between male and female sharks in the ocean [105,106] can lead to sex differences in ecological and demographic parameters [107,108,109,110]. Moreover, life history parameters such as longevity and mortality demonstrate sex differences. Therefore, a two-sex stage-based population model, rather than the conventional single-sex assessment model, was considered appropriate for the current study. Moreover, accessing multiple paternity as a mating strategy because of region-specific movement behavior suggested that mako sharks have significantly more opportunities to meet mating partners than previously believed, potentially indicating the existence of polyandry [45], contributing to improved demographic models. In addition, when formulating management measures, a species’ possible sexual dimorphism must be carefully evaluated to determine whether more accurate estimates of decline risk can be derived.

### 4.3. Conservation Problems and Management Strategies for Mako Sharks

The current demographic analysis results demonstrate that fishing negatively affects the mako shark population, and for the first time, provided data for implementing mako shark management and conservation in the South Pacific Ocean using a demographic method. Studies have indicated that mako shark communities in other regions experience considerable changes: the Mediterranean Sea stock has collapsed and is at the risk of extinction [6], the North Atlantic stock remains overfished [111], and the status of the South Atlantic stock remains undetermined. The most recent comprehensive studies on the North Pacific and Indian Ocean stocks have indicated that the populations were neither overfished nor stable [22,112].

The current conclusions are based on a two-sex stage-structured matrix population model, and stochastic approaches for parameters with uncertainty may provide a relatively realistic perspective of the population dynamics of this species. Our results revealed that mako shark populations would only be able to sustain a relatively low fishing mortality (20% of natural mortality) in the study area. Little research has focused on quantitative stock assessment or fishery status indicators. Most researchers have used a few remarkable methods for including *M_s_* into their demographic models. Several slow-growing, long-living shark species have demonstrated substantially lower *λ* [52]. When we incorporated *F* into our demographic analysis, the population growth rates of mako sharks decreased. This result also showed that any added anthropogenic source of mortality will likely result in its population decline.

In this study, the effects of protection (Figure 7) differed according to the sex-specific hypothesis. In both reproductive cycle models, the protection of immature mako sharks of both sexes resulted in a higher population growth rate than the protection of mature sharks. Shark demographic studies have reported that the population growth rate is the most sensitive to the immaturity stage; it is particularly high in juvenile females [30,31,38,79]. The differential availability of shark sexes in fisheries possibly arises from sex segregation in the population [83]; moreover, the mako shark demonstrates substantial sex and size segregation as well as segregation between juveniles and other developmental stages. Therefore, protection of this species using an optimal management approach dependent on the varying sex-specific hypothesis can be the most efficient measure for mako shark conservation. However, as ocean boundaries between male and female sharks have not been documented in detail, the potential role of sexual segregation in the overexploitation of shark populations remains unclear [83], particularly in the current study, considering that the knowledge on the South Pacific mako shark population is limited. A population with a clear structure and a 2-year reproductive cycle was noted to remain stable, whereas that with a longer (3-year) reproductive cycle was predicted to diminish.

### 4.4. Life History Parameter Uncertainty and Their Limitation

This study considered not only the estimated parameters but also the CIs to reflect and account for these uncertainties. For this, we applied Monte Carlo simulations for the survival and fecundity parameters input in the Leslie matrices. The results of the demographic analysis using stage-based matrix models in this study should be interpreted with caution because the analysis was based on some restricted assumptions. The biological and life history parameter data of South Pacific mako sharks are possibly influenced by bias. Our analysis was based only on the relevant biological parameters derived for mako sharks in the Southeastern Pacific Ocean (Cerna and Licandeo [27]). Moreover, the life history data reported by Bishop et al. [26] in the Southwestern Pacific Ocean were also used in this study. More robust growth curves for improved stock assessments and demographic analyses are warranted.

Stage-based models have been modified based on the life history parameters of different species and have been efficiently applied in several studies including those on the sandbar shark (*Carcharhinus plumbeus*) [54], spiny dogfish (*Squalus suckleyi*) [42], mako shark [19,20,45], silky shark (*Carcharhinus falciformis*) [32], and pelagic thresher shark (*Alopias pelagicus*) [31]. In this study, rather than adopting an age-based approach, we used a stage-based matrix, and simulations revealed that a 2-year reproductive cycle predicted more optimistic results than a 3-year reproductive cycle. Hence, the existence of a resting stage significantly affects population growth rates. However, because of the large size and wide oceanic distribution of mature female sharks as well as high fishing activity, determining the reproductive and life history parameters of mako sharks is difficult [10,83,113].

Age at maturity is a crucial parameter potentially influencing the demographic analysis results [52,59,75]. In this study, we used a stochastic methodology to account for annual fluctuations in age at maturity in the models and improved the quality of the results. Female maturation age has been reported to have the greatest effect on the mako shark population growth rate [53]. This study demonstrated the importance of considering the sex differences in age at maturity when estimating the population growth rates, with extreme differences in life history parameters—corroborating the results of previous studies [19,32,52,114,115]. Fecundity, one of the most important productivity indicators, is uncertain in most cases; this relationship has not been confirmed for the litter size and the size of female mako sharks [8,9,10,12,76,77,78]. Therefore, rather than lognormal or uniform assumed distributions, we used a triangular distribution in this study to determine the fecundity parameters and assign the most likely value between the limits. Further research into the reproductive traits of this species is required to effectively assess its stock. Future studies should focus on collecting more reliable litter size data through onboard observation to reduce uncertainty.

The unavailability of longevity data was also a potential source of bias influencing our demographic analysis results. Obtaining biological data including information on the longevity of a species is difficult, and the lack of these data has hindered the demographic analysis of shark populations [39]. Almost all sharks are bycatch species with large body sizes, small populations, high susceptibility, and widespread distribution. The oldest fish may be caught in some cases. As obtaining shark longevity data through direct age verification is difficult, longevity is typically calculated through indirect analysis. The current study incorporated stochastic methodology to determine the longevity by using four equations reported previously [79,80,81]. This approach may reflect the actual conditions more closely. Simultaneously, the periodicity of band pair deposition may vary at the different life stages of mako sharks [116], leading to age estimation uncertainties. Furthermore, because the observed maximum age reported in the South Pacific Ocean was smaller than that in other oceans, longevity may have been underestimated. Nevertheless, we did not use the observed longevity in this study.

Compensatory density dependence is vital for fisheries management because they can offset the loss of individuals [117,118]. Although the concept of compensation is simple, it is one of the most contentious topics in population dynamics [118,119,120,121]. These responses may be smaller in shark species [19,32,54], and they are particularly unlikely to influence mako sharks due to their extremely susceptible life cycle characteristics (i.e., long lifespan, low fecundity, late maturity, and an extended reproductive cycle), all of which make them more susceptible to the overexploitation of high fishing pressure. Although a sustainable harvest is possible in the absence of compensating density dependency in theory, it is ecologically impractical [118]. Nevertheless, in this study, when *F* was introduced and the demographic model did not consider density dependency or compensation, mako shark stock status demonstrated a decline (Figure 6). This result is similar to those of Tribuzio and Kruse [42], Tsai et al. [18,31,32], Coelho et al. [115], and Chang and Liu [20]. Therefore, even though we ignored the compensation of density reliance in the South Pacific mako shark population, the current results could be considered as acceptable. Demographic models provide certain theoretical and empirical results that facilitate future advancements in compensating process measurements and understanding.

The estimations of stage-based models in this study assumed that mako sharks in the South Pacific form a unit stock. According to Heist et al. [122] and Schrey and Heist [123], a unit stock of mako sharks was confirmed in the eastern Pacific Ocean across the equator. The data derived using molecular genetic techniques have recently revealed at least three genetic populations of mako sharks in the Pacific Ocean including the North Pacific, Southeast Pacific, and Southwestern Pacific Ocean, with little movement of fish between the populations [124]. Moreover, any changes in the life history parameters in demographics influence the population dynamics [125], with the focus being on the population growth rate. Life history parameters can be input into species-specific demographic models based on unit stock. However, the South Pacific Ocean mako shark stock structure is unclear, mostly due to the lack of relevant data. Clarifying whether mako sharks in the entire South Pacific Ocean are a unit stock [124] is necessary to establish a completely separate demographic model for each stock and to gain a further understanding of the population dynamics of the species.

Additionally, because of the parameters used, the accuracy of the information, and the large variability degree of uncertainty, it is impossible to recommend a single value, demonstrating that the indirect estimation of *M* is less accurate [64]. Notably, Zhou et al. [126] indicated concerns regarding the estimation of *M* for elasmobranchs (including sharks) by using estimators generated mostly from teleost data. Therefore, in this study, we emphasized the uncertainty of *M* estimates for stock assessment and management [63], particularly in demographic models. Actual variations in *M* may be much higher than assumed in this study. Thus, a robust method for estimating *M* by using tagging research findings and catch and effort data and thus calculating the total mortality rate is required.

## 5. Conclusions

Even though mako sharks are vulnerable to overexploitation, management strategies have not been established for this species thus far. Here, we examined mako shark demographics in the South Pacific Ocean by using limited data on their life history. Nevertheless, our sex-specific demographic models demonstrated that management decisions in the South Pacific Ocean mako shark population are required to ensure the sustainability of mako shark stocks, and that the population growth rate is higher when immature mako sharks are protected than when mature mako sharks are protected. Better estimates of natural and fishing mortality are required to understand the impact of commercial fisheries in the South Pacific Ocean’s shortfin mako shark population. It is also suggested that our approach can be utilized as an assessment tool for shark species with insufficient catch and effort data.

## Figures and Tables

**Figure 1 animals-12-03229-f001:**
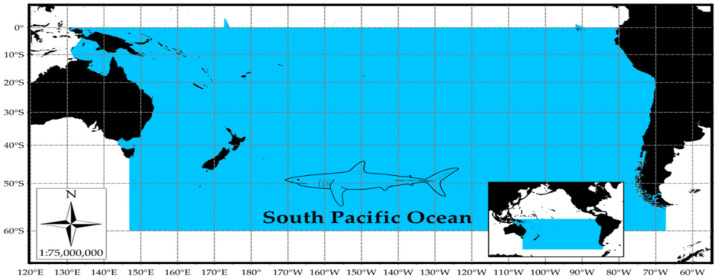
Map of the study area. The blue area depicts the South Pacific Ocean’s location on the world map.

**Figure 2 animals-12-03229-f002:**
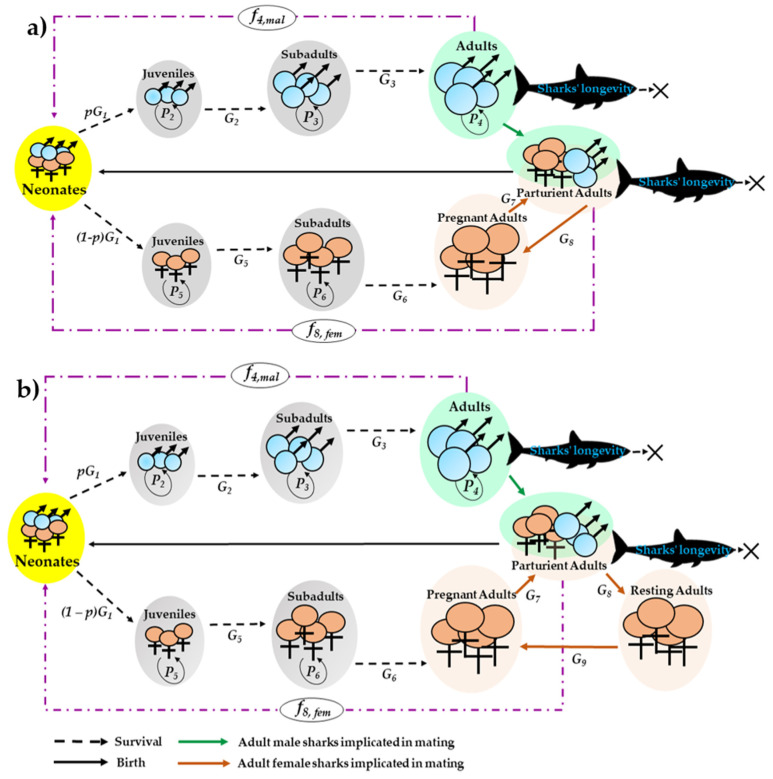
Stage-based matrix model for 2-year (**a**) and 3-year (**b**) reproductive cycles.

**Figure 3 animals-12-03229-f003:**
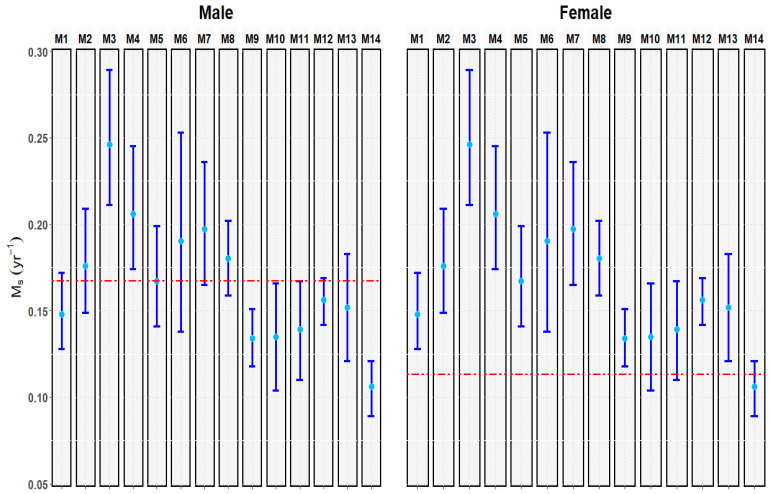
*M_s_* of male and female mako sharks in the South Pacific Ocean. The lower and upper values are delineated by vertical navy blue lines, and each azure dot reflects the median values of each estimator, respectively. The red dashed line in each graph represents the mean *M_s_* of the 14 estimators.

**Figure 4 animals-12-03229-f004:**
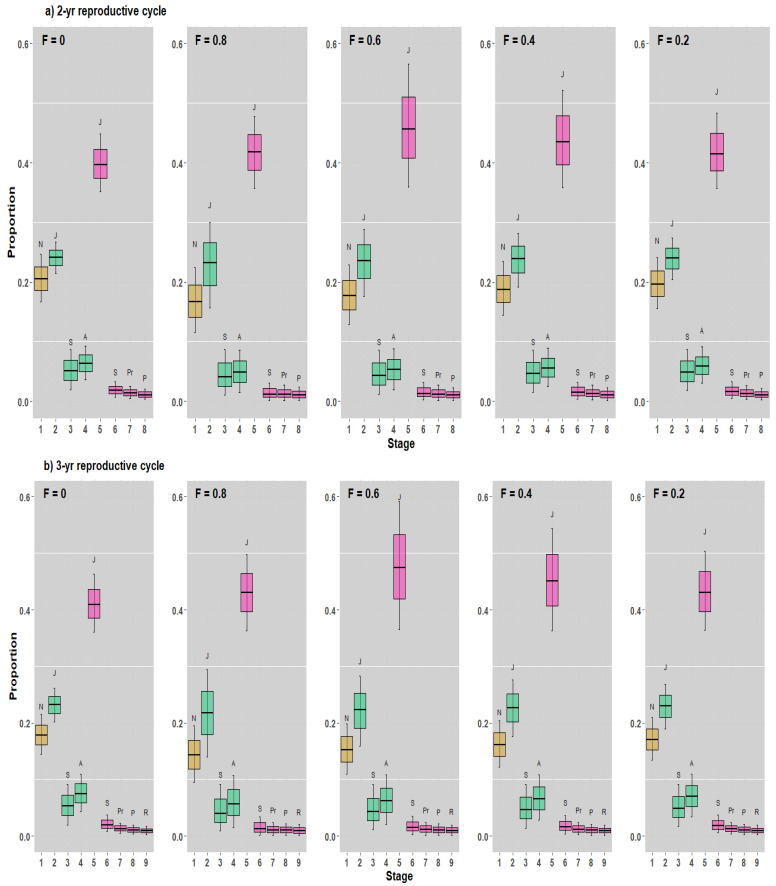
SSD values at various levels at F = 0, 0.8, 0.6, 0.4, and 0.2. (**a**,**b**) SSD values in the 2- and 3-year reproductive cycle models, respectively (where N = neonates, J = juveniles, S = subadults, A = adults, Pr = pregnant adults, P = parturient adults, R = resting adults). Both sexes (male [in aquamarine] and female [in hotpink]) are shown.

**Figure 5 animals-12-03229-f005:**
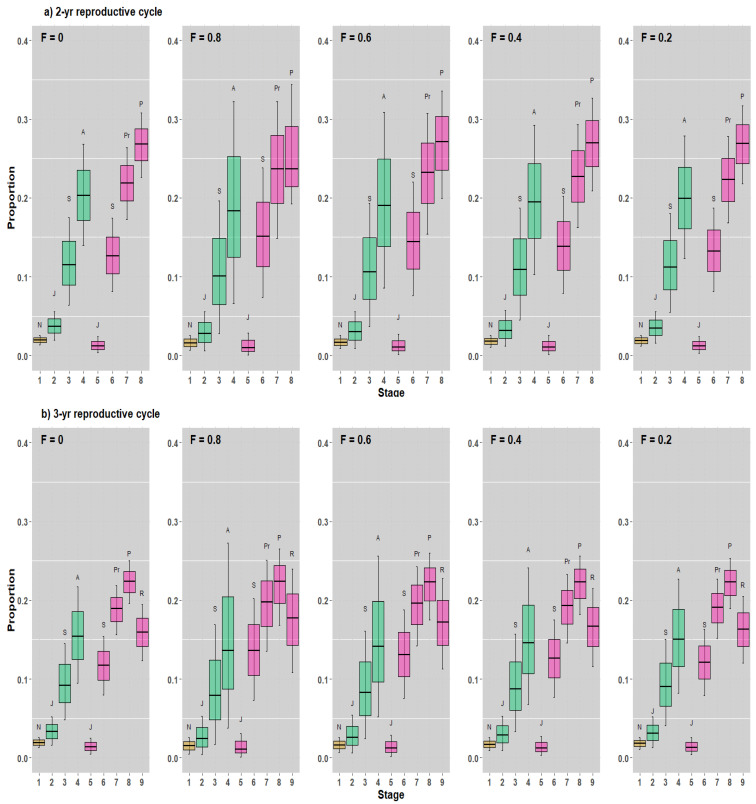
RV values at various levels at F = 0, 0.8, 0.6, 0.4, and 0.2. (**a**,**b**): RVs in the 2- and 3-year reproductive cycle models, respectively (where N = neonates, J = juveniles, S = subadults, A = adults, Pr = pregnant adults, P = parturient adults, R = resting adults). Both sexes (male [in aquamarine] and female [in hotpink]) are shown.

**Figure 6 animals-12-03229-f006:**
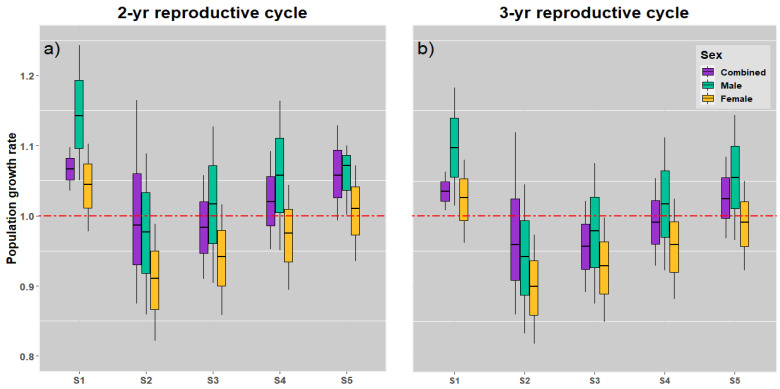
Box plots for the mako shark population growth rate in various harvesting scenarios with 2-year (**a**) and 3-year (**b**) reproduction cycle models. The red dashed line represents the stable level of population growth.

**Figure 7 animals-12-03229-f007:**
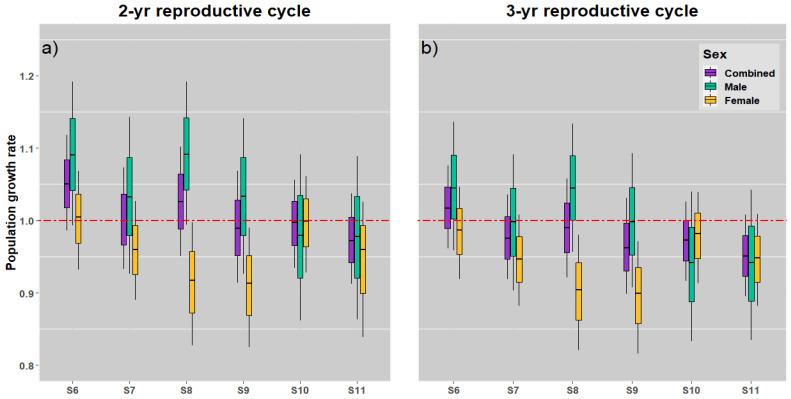
Box plots for mako shark population growth rate in various stock protection scenarios with 2-year (**a**) and 3-year (**b**) reproductive cycle models. The red dashed line represents the stable level of population growth.

**Table 1 animals-12-03229-t001:** Biological parameters of mako sharks used in this study.

	L_∞_ (cm)	K (Year^−1^)	t_0_ (Years)	L_0_ (cm)	Reference
Male	296.604	0.087	−3.579	79.3	[52]
Female	325.293	0.076	−3.182	70.0	[52]

L_∞_: asymptotic length, k: growth coefficient, t_0_: age at length 0, L_0_: length at birth.

**Table 2 animals-12-03229-t002:** Life history stages of mako sharks in the South Pacific Ocean.

Sex	Stage-Class	Approximate Ages (Years)	Expected Stage Duration (Years)
Male	Neonates	0–1	1
	Juveniles	1–6	5
	Subadults	6–a*_mat_*	1–4
	Adults	a*_mat_*–a*_max_*	12–24
Female	Neonates	0–1	1
	Juveniles	1–17	16
	Subadults	17–a*_mat_*	2–4
	Adults	a*_mat_*–a*_max_*	15–26
Included	Pregnant adults	>21	1
	Parturient adults	>21	1
	Resting adults *	>21	1

a*_mat_*: 7–10 years for males and 19–21 years for females, a*_max_*: 22–31 years for males and 36–45 years for females, * Only related to the 3-year reproductive cycle.

**Table 3 animals-12-03229-t003:** Method and formulas used to estimate *M_s_* of mako sharks in the South Pacific Ocean.

ID	Method	Equation
M1	Campana et al. [66]	$M_{s}=-\frac{\ln(0.01)}{a_{\max}}$
M2	Hoeing [67]	$M_{s}=0.941-0.873\ln(a_{\max})$
M3	Then et al. [68]	$M_{s}=4.899a_{\max}^{-0.916}$
M3	Then et al. [68]	$\ln(M_{s})=1.717-1.01\times \ln(a_{\max})$
M5	Hamel [69]	$M_{s}=\frac{4.374}{a_{\max}}$
M6	Zhang and Megrey [70]	$M_{s}=\frac{3k}{e^{k0.38a_{\max}}-1}$
M7	Jensen [71]	$M_{s}=\frac{1.65}{a_{mat}}$
M8	Hisano et al. [41]	$M_{s}=\frac{1.6}{a_{mat}-t_{0}}$
M9	Frisk et al. [72]	$M_{s}=\frac{1}{0.44a_{mat}+1.87}$
M10	Cubillos et al. [73]	$M_{s}=4.31{\left[t_{0}-\frac{\ln(0.05)}{k}\right]}^{-1.01}$
M11	Jensen [71]	$M_{s}=1.6k$
M12	Frisk et al. [72]	$\ln(M_{s})=0.42\ln(k)-0.83$
M13	Hamel [69]	$M_{s}=1.753k$
M14	Then et al. [68]	$M_{s}=4.118k^{0.73}L_{\infty }^{-0.33}$

*a_max_* = longevity, set at 22–31 years for male sharks and 36–45 years for female sharks, *a_mat_* = age at maturity, set at 7–10 years for male sharks and 19–21 for female sharks, *L_∞_*, *t*_0_ and *k* = growth parameters of the Von Bertalanffy Growth Equation (VBGE) data (Table 1).

**Table 4 animals-12-03229-t004:** Uncertainty used in the stochastic simulations.

Sources of Uncertainty	Male	Female	Assumed Distribution
Fecundity	Triangle (8, 12.5, 18)	Triangle (8, 12.5, 18)	Triangular distribution
Natural mortality	ln (mean, SD) *	ln (mean, SD) *	Lognormal
Age at maturity	7–10 years	19–21 years	Uniform
Longevity	22–31 years	36–45 years	Uniform

* Mean and (SD) obtained for *M_s_* across 14 approaches for each sex and stage used to establish a lognormal distribution.

**Table 5 animals-12-03229-t005:** Comparison of mako shark population growth rates collected from different studies.

Area	Model/Method	Sex	*a_mat_*	*a_max_*	*λ* (yr^−1^)	Reference
Northwestern Atlantic	Age-structured matrix population model	F	6, 7, 8	17	1.141 ^a^ (1.098–1.181) ^t^	[52]
Atlantic	Age-structured matrix population model	F	6, 7, 8	17	* 1.076 ^a^	[94]
North Atlantic	Bayesian surplus production model	F	18	32	* 1.060–1.061 ^y^	[95]
South Atlantic	Bayesian surplus production model	F	18	32	* 1.060–1.064 ^y^	[95]
North Pacific	Two-sex stage structured matrix population model	C	-	-	1.078 ^b(2yr)^, 1.051 ^b(3yr)^	[19]
		M	11–15	24–31		
		F	18–21	31–41		
North Pacific	Two-sex stage-structured matrix population model with polyandrous and polygynous	C	-	-	1.010–1.082 ^c,y^	[45]
		M	11–15	24–31		
		F	18–21	31–41		
Global	Two-sex age-structured matrix population model	C	-	-	* 1.107 ^d^ (1.007–1.374) ^x^	[53]
		M	3	23		
		F	17	38		
Northwestern Pacific	Stage-based models	F	19–22	42	1.059 ^e^ (1.023–1.098) ^z^	[20]
Indian Ocean	Two-sex age-structured matrix population model	C	-	-	* 1.120 ^f^ (1.062–1.141) ^x^	[21]
		M	7	23–29		
		F	15	32–38		
South Pacific	Two-sex stage-structured matrix population model	C	-	-	1.098 ^g (2yr)^ (1.036–1.165) ^t^ 1.063 ^g (3yr)^ (1.008–1.119) ^t^	This study
		M	7–10	22–31		
		F	19–21	36–45		

C: Combined, M: Male, F: Female, *a_mat_*: age at maturity, *a_max_*: longevity, ^2yr and 3yr^: 2- and 3-year reproductive cycle models, respectively, ^x^ Minimum to maximum population growth rates, ^y^ Population growth rate estimated in several scenarios, ^z^ 95% CI, ^t^ 95% CI, calculated as the 2.5th and 97.5th percentiles, * *r* of the matrix population model, converted to, *λ*^a^ Estimated using six empirical methods: (1) Hoenig [67], (2) Pauly [102], (3) Chen and Watanabe [103], (4) Peterson and Wroblewski [104], and (5) and (6) Jensen [71]. ^b^ Estimated using life history invariants of (1) Hoenig [67], (2) Campana et al. [66], (3) and (4) Jensen [71], and (5) Petersen and Wroblewski [104]. ^c^ Estimated using the methods of (1) Hoenig [67], (2) Jensen [71], (3) Peterson and Wroblewski [104], and (4) Campana et al. [66]. ^d^ Estimated using several theoretical models: (1) Chen and Watanabe [103], (2) Jensen [71], (3) Pauly [102], (4) Hoenig [67], and (5) Petersen and Wroblewski [104]. ^e^ Estimated by randomly selecting from the following four methods: (1) Hoenig [67], (2) and (3) Jensen [71], and (4) Petersen and Wroblewski [104]. ^f^ Estimated by randomly selecting from the following methods: (1) Petersen and Wroblewski [104] and (2) Hoenig [67]. ^g^ Estimated by randomly selecting from the following 14 methods: (1) Hoeing [67], (2) Campana et al. [66], (3–5) Then et al. [68], (6–7) Hamel [69], (8) Zhang and Megrey [70], (9–10) Jensen [71], (11–12) Frisk et al. [72], (13) Hisano et al. [41], and (14) Cubillos et al. [73].

## Data Availability

Not applicable.

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
