# Peer review of "Demographic Analysis of Shortfin Mako Shark (Isurus oxyrinchus) in the South Pacific Ocean"

_animals, 2022, doi:10.3390/ani12223229_

Round 1
Reviewer 1 Report
Major Comments:
Firstly, I would like to thank the authors as I found this an engaging and enjoyable manuscript to read. The authors presented a preliminary analysis on the stock of Mako Sharks in the South Pacific. They modelling multiple different life stage parameters to create simulations of important population dynamics, which are relevant to predict stocks. Using nested models including 2 vs 3 year life cycles, male vs female vs combined effects as well as different levels of Fishing Pressure (F) across the whole population and the different sexes and demographic life stages of those sexes. The authors highlighted many points of potential bias within their methods and the lack of available landings/bycatch data.
The main correction I feel is necessary for this paper is to move some of the tables to supplementary materials, especially when the data themselves are present within figures and easier for the reader to digest. I would suggest that tables 5, 6, 7 and 8 aren't necessary within the main manuscript. I have commented on figures 4,5, 6 and 7 below and feel once these graphs are made clearer to the reader and made more consistent with colour legends and similar graph layouts (See comments below) the need for the tables disappears and makes the manuscript easier to read.
Minor Comments:
Line 50: add "to" before repot.
Line 56: add "International Union for Conservation of Nature (IUCN)" before "Red List" (Granted it is a well known acronym but still necessary).
Line 56: likewise CITES is an acronym and should be stated in full for its first use. (Convention on International Trade in Endangered Species of Wild Fauna and Flora).
Line 59: remove "becoming". The characteristics aren't changing so the vulnerability isn't changing. Therefore, it is extremely susceptible to overexploitation.
Line 66-69: "Mako shark stock status in the South Pacific Ocean, however, remains unclear due to the lack of quantitative complete stock assessments and significant uncertainty in both fisheries and biological information or limited data available for the studied populations." This sentence is very hard to read. Perhaps change to: "However, Mako shark stock status in the South Pacific Ocean remains unclear due to significant uncertainty in both fisheries and biological information as well as the lack of quantitative stock assessments for the studied populations."
Line 79: remove comma after "[19, 20]," and replace with "and".
Figure 1: x and y axis tick labels have lots of unnecessary zeros, which likely has lead to these labels being small and harder to read, for such large areas of the globe fewer, more spread out and larger axis tick labels would be more aesthetically pleasing.
Line 180: is fi meant to be like within the equations with subscript i after function symbol?
Equations: generally some of the equations look distorted or flattened. (Might be the viewing process I am using though).
Lines 226-227: surely M is natural mortality and therefore isn't related to exploitation or management?
Line 240: change "In the many" to "As with many".
Line 240: remove "considering that"
Line 260: change "lowest" to "lower".
Table 3: First use of the acronym VBGE, please define at first use.
Line 268: use of inf and symbol for infinite (do they mean different things?).
Line 279: replace "Because" with "As"
Lines 288-305: remove use of "equals" in each bullet point as unnecessary.
Line 329: "increasing population increase" is odd. Remove "increase" after "popultation".
Line 330: remove "(2.5th and 97.5th Percentiles)". This was already established as your 95% CI in the methods.
Line 348-350: I think "underestimated" is wrong, this implies the values for the 3-year cycle model are incorrect but you haven't directly tested this. Maybe replace with "estimated a lower". This would read as follows: "In summary, the aforementioned findings confirmed that the 3-year reproductive cycle model estimated a lower population growth rate compared with the 2-year reproductive cycle, particularly in female sharks.".
Line 355: Perhaps restate the SSD and RV acronyms. (They were defined in the methods but might make easier for the reader).
Figure 4: axis titles are cut off in the top graph.
Figure 4: I am finding working out what each box in the plots represents. Some boxes have specific labels (J-M etc.). However, other boxes don't have labels. There is colouring for the boxes but no legend. A legend for colours could make it easier to distinguish which stages are which and which ones are sex specific and which aren't.
Figure 5: as with Figure 4, hard to work out the stages and colours used for some boxes but not others. A legend for boxes and their colours, perhaps consistently across figures 4 and 5 would be useful.
Figure 6: the legend for colours is the same for 2 year and 3 year reproduction so can just be displayed once. It seems as you are using ggplot2 you could make use of the patchwork package to display the graphs side by side. This package allows you to collect guides. (Not necessary, just a useful package I use, I know there are lots to chose from).
Figure 7: I feel Figure 7could also have cases on the x axis like figure 6, rather than using facetting by case. This would make it easier to compare within the reproductive cycles. Plus, as above, similar comment on one colour legend for both graphs within the figure.
Line 447: remove "variability"
Line 491: add "than" between "higher" and "those".
Line 491-492: replace "that higher" with "values greater".
Line 576-577: replace "between" with "for".
Line 588: replace "Because" with "As"
Line 606: i suggest rearranging to say what the life characteristics are mean the sharks are susceptible to. e.g. "to how their life cycle characteristics make them extremely susceptible to..." then what they are susceptible to e.g. "overexploitation/fishing pressure.".
Line 638: remove "is".
Line 644: change "lower" to "relatively low".
Line 667: replace "because" with "as".
Author Response
Responses Reviewer 1's Comments
Major Comments:
Firstly, I would like to thank the authors as I found this an engaging and enjoyable manuscript to read. The authors presented a preliminary analysis on the stock of Mako Sharks in the South Pacific. They modelling multiple different life stage parameters to create simulations of important population dynamics, which are relevant to predict stocks. Using nested models including 2 vs 3 year life cycles, male vs female vs combined effects as well as different levels of Fishing Pressure (F) across the whole population and the different sexes and demographic life stages of those sexes. The authors highlighted many points of potential bias within their methods and the lack of available landings/bycatch data.
The main correction I feel is necessary for this paper is to move some of the tables to supplementary materials, especially when the data themselves are present within figures and easier for the reader to digest. I would suggest that tables 5, 6, 7 and 8 aren't necessary within the main manuscript. I have commented on figures 4, 5, 6 and 7 below and feel once these graphs are made clearer to the reader and made more consistent with color legends and similar graph layouts (See comments below) the need for the tables disappears and makes the manuscript easier to read.
Response:
We acknowledge reviewer’s comments and have revised the tables 5, 6, 7, and 8 as suggested in current version. In addition, the remaining minor corrections in the electronic copy have also been made accordingly.
Minor Comments:
Line 50: add "to" before repot
Response: This MS has been added “to” as suggested.
Line 56: add "International Union for Conservation of Nature (IUCN)" before "Red List" (Granted it is a well known acronym but still necessary).
Response: This MS has been added "International Union for Conservation of Nature (IUCN)" as suggested.
Line 56: likewise CITES is an acronym and should be stated in full for its first use. (Convention on International Trade in Endangered Species of Wild Fauna and Flora).
Response: Revised as suggested in the current version.
Line 59: remove "becoming". The characteristics aren't changing so the vulnerability isn't changing. Therefore, it is extremely susceptible to overexploitation.
Response: This MS has been removed “becoming” as suggested.
Line 66-69: "Mako shark stock status in the South Pacific Ocean, however, remains unclear due to the lack of quantitative complete stock assessments and significant uncertainty in both fisheries and biological information or limited data available for the studied populations." This sentence is very hard to read. Perhaps change to: "However, Mako shark stock status in the South Pacific Ocean remains unclear due to significant uncertainty in both fisheries and biological information as well as the lack of quantitative stock assessments for the studied populations."
Response: The sentence was changed as suggested. See as below:
"However, mako shark stock status in the South Pacific Ocean remains unclear due to significant uncertainty in both fisheries and biological information as well as the lack of quantitative stock assessments for the studied populations."
Line 79: remove comma after "[19, 20]," and replace with "and".
Response: Revised as suggested.
Figure 1: x and y axis tick labels have lots of unnecessary zeros, which likely has lead to these labels being small and harder to read, for such large areas of the globe fewer, more spread out and larger axis tick labels would be more aesthetically pleasing.
Response: Figure 1 has been changed as suggested.
Line 180: is fi meant to be like within the equations with subscript i after function symbol?
Response: Revised as suggested with subscript i
Equations: generally some of the equations look distorted or flattened. (Might be the viewing process I am using though).
Response: All equations in this MS have been modified to fit in the current version as suggested.
Lines 226-227: surely M is natural mortality and therefore isn't related to exploitation or management?
Response: Natural mortality (M) is commonly considered one of the most important parameters in a fish stock assessment, influencing population productivity estimates. M, in other words, is a significant predictor of productivity in population models and one of the most influential factors in fisheries stock assessment and management. Therefore, M is closely related to exploitation or management.
Line 240: change "In the many" to "As with many".
Response: The wording has been changed as suggested.
Line 240: remove "considering that"
Response: The sentence has been removed “considering that” as suggested.
Line 260: change "lowest" to "lower".
Response: The wording has been changed to “lower” as suggested.
Table 3: First use of the acronym VBGE, please define at first use.
Response: Table 3 has been defined of the acronym VBGE as suggested.
Line 268: use of inf and symbol for infinite (do they mean different things?).
Response: Revised as suggested, Linf = L¥, which is the equivalent symbol for infinite.
Line 279: replace "Because" with "As"
Response: The wording has been replaced as suggested.
Lines 288-305: remove use of "equals" in each bullet point as unnecessary.
Response: This entire section has been removed “equals” as suggested.
Line 329: "increasing population increase" is odd. Remove "increase" after "popultation".
Response: This sentence has been removed “increase” as suggested.
Line 330: remove "(2.5th and 97.5th Percentiles)". This was already established as your 95% CI in the methods.
Response: This sentence has been removed “(2.5th and 97.5th Percentiles)” as suggested.
Line 348-350: I think "underestimated" is wrong, this implies the values for the 3-year cycle model are incorrect but you haven't directly tested this. Maybe replace with "estimated a lower". This would read as follows: "In summary, the aforementioned findings confirmed that the 3-year reproductive cycle model estimated a lower population growth rate compared with the 2-year reproductive cycle, particularly in female sharks.".
Response: We agree with reviewer’s comment. This sentence has been replaced by “estimated a lower” in the current version.
Figure 4: axis titles are cut off in the top graph.
Response: Figure 4 has been changed as suggested.
Figure 4: I am finding working out what each box in the plots represents. Some boxes have specific labels (J-M etc.). However, other boxes don't have labels. There is colouring for the boxes but no legend. A legend for colours could make it easier to distinguish which stages are which and which ones are sex specific and which aren't.
Response: Figure 4 has been changed as suggested.
Figure 5: as with Figure 4, hard to work out the stages and colours used for some boxes but not others. A legend for boxes and their colours, perhaps consistently across figures 4 and 5 would be useful.
Response: This has been changed Figure 5 as suggested.
Figure 6: the legend for colours is the same for 2 year and 3 year reproduction so can just be displayed once. It seems as you are using ggplot2 you could make use of the patchwork package to display the graphs side by side. This package allows you to collect guides. (Not necessary, just a useful package I use, I know there are lots to chose from).
Response: This has been changed Figure 6 as suggested.
Figure 7: I feel Figure 7 could also have cases on the x axis like figure 6, rather than using facetting by case. This would make it easier to compare within the reproductive cycles. Plus, as above, similar comment on one colour legend for both graphs within the figure.
Response: This has been changed Figure 7 as suggested.
Line 447: remove "variability"
Response: This MS has been removed “variability” as suggested.
Line 491: add "than" between "higher" and "those".
Response: This MS has been added “than” as suggested.
Line 491-492: replace "that higher" with "values greater".
Response: The wording has been replaced as suggested.
Line 576-577: replace "between" with "for".
Response: The wording has been replaced with “for” as suggested.
Line 588: replace "Because" with "As"
Response: The wording has been replaced as suggested.
Line 606: I suggest rearranging to say what the life characteristics are mean the sharks are susceptible to. e.g. "to how their life cycle characteristics make them extremely susceptible to..." then what they are susceptible to e.g. "overexploitation/fishing pressure.".
Response: Revised as suggested. See as below
“These responses may be smaller in shark species [19,27,53], and they are particularly un-likely to influence mako sharks due to their extremely susceptible life cycle characteristics (i.e., long lifespan, low fecundity, late maturity, and an extended reproductive cycle), all of which make them more susceptible to overexploitation of high fishing pressure.”
Line 638: remove "is".
Response: This MS has been removed “is” as suggested.
Line 644: change "lower" to "relatively low".
Response: The wording has been changed to “relatively” as suggested.
Line 667: replace "because" with "as".
Response: The wording has been replaced with “as” as suggested.

Reviewer 2 Report
The manuscript is a sound paper, well presented, dealing with an interesting approach.
Author Response
Responses Reviewer 2's Comments
Comments and Suggestions for Authors:
The manuscript is a sound paper, well presented, dealing with an interesting approach.
Response: We acknowledged the reviewer’s support of our approach in this MS.

Reviewer 3 Report
I would like to commend the authors on this very interesting and important study. Understanding population dynamics and reliably predicting the effects of fishing pressure on different life history stages and sexes of different species will be key to developing sound management strategies and ensuring sustainable harvests. The following comments I made should be seen as suggestions.
Major comments:
Figures
Overall, I suggest you increase the in-graph font size to improve readability. The text on some graphs is impossible to read. I provide further details in the comments below.
Discussion
I feel this is the part that needs the most work. You tend to repeatedly list the limitations of your study and the difficulties in assessing the stock of mako sharks in the South Pacific Ocean. I recommend you move those issues to a single section of the discussion. Also, avoid being too negative about the results of your study. Yes, your study has some limitations, every study does. But be sure not to make it sound as if the results are completely unreliable and that what I just read is nothing but a preliminary pilot study. You may lose your reader’s attention. This is also the reason why I suggest you move the section about the limitations of your study to the end of the discussion. Make sure this section isn’t too long. You could easily shorten it a bit by removing the parts on the types of models you used and the data you used. This should all be in the methods and there is no need repeating it all in the discussion. If part of what you wrote in the discussion is missing in the methods section, make sure you add it there. Instead of explaining what you did again, try to focus on the implications these biases and limitations may have and how we could reduce these uncertainties in the future. How did other researchers deal with the limitations of their studies?
Overall, I think you need to interpret your results a bit more from an ecological perspective. You mentioned mako shark stocks that are almost extinct. Have there been any models predicting the population decline? How accurate were those estimates? How could your model help predict population dynamics of mako sharks in the future? Could your model be adjusted to fit mako shark populations elsewhere or even different species? Your model predicts that protecting sub-adults will have a greater impact on the population growth rate than protecting adults. What does this mean for management? Are mako sharks mostly bycatch or is there a fishery targeting this species?
Minor comments:
Simple Summary
Line 10 – Statistical data of what kind? Please be more specific.
Line 16 – “…fishery data.”
Abstract
Line 31 – 33 – This sentence needs to be reworded. Firstly, you use the term demonstrate twice, so I would suggest you use a synonym for one of them. Secondly, to make the conclusive statement a bit easier to follow, I suggest you state “…reproductive cycle compared to the 3-year reproductive cycle.”.
Introduction
Line 50 – “…(IATTC) has yet to report its shark stock assessment…”
Line 54 – consider changing it to “…fast-swimming species, and a vital component of…”
Line 57 – 58 – I am missing the conclusion of this sentence. You state that I. oxyrinchus has several life history characteristics, then list them. But doesn’t every species have several life history characteristics? In the next sentence, you are pointing out the consequences of the mako shark’s long life, late maturity, late fecundity, and extended reproductive cycle. I suggest you either combine the two sentences into one, which would probably make it quite lengthy, or you change the structure of this sentence, so it simply introduces the characteristics. For example: “This species is known for its long life, low fecundity, late maturity, and extended reproductive cycle (2-3 years).”
Line 59 – You started the previous sentence with “This species…”. Using the term again feels repetitive. Maybe use the species name instead. You could also refer to it as “it” since this statement follows up on the previous sentence.
Line 66 – 69 – Do you have any reference for this statement? Any previous stock assessments or reports on the uncertainty in fisheries and biological information?
Line 69 – 70 – Delete this sentence. You are still providing a background for your study. You have introduced the knowledge gap. Now you are providing the methods that may be used to close that gap with examples. Your study, including your motivation to do it, should come last.
Line 70 – Remove “However,…”. Start the sentence with “Obtaining the long term…”
Line 83 – 88 – You should provide some references for these three sentences.
Line 95 – Please provide the scientific names for boarfish, electric knifefish, and European sea bass.
Materials and Methods
Figure 1 – The text within the graph is too small to read. Please increase the font size of the scale a bit.
Figure 2 – You should increase the font size of the in-graph text a bit. At least the text on the purple dotted lines, as it is not readable.
Table 2 – If the gestation period is two years, why is the expected duration of the pregnant adult stage only one year? Also, if parturient, by definition, refers to the stage of giving birth, an expected duration of one year appears to be very long. My question, in this case, would be if this duration was chosen due to the nature of the model you used. Furthermore, I wonder if you could include parturient females in the pregnant stage.
Line 153 – What is an “ungenerous technique”?
Line 154 – What do you mean by “allowed injuries”?
Line 174 – “…in the 2-year reproductive cycle…”
Equation 6 – The numbering for this equation is missing. Also, the subscript and superscript text is very difficult and partly impossible to read. Consider increasing the font size slightly for the equations to increase readability.
Line 227 – 228 – How is the natural mortality rate closely related to exploitation rates and management variables?
Line 288 – “…was set to equal 80%...” – change this throughout the list of scenarios.
Results
Line 314 – “…values than female sharks…”
Figure 3 – Please provide more information in the figure caption. What does each orange dot represent? What do the vertical blue lines show? Given there is only one dashed red line for each sex, does it represent the overall mean natural mortality value across all 14 estimators? Also, please increase the font size of the in-graph text a bit to improve the readability. I also suggest you change the colour of the 14 estimators. Black text on a dark background makes it unreadable now.
Line 329 – What do you mean by “…an increasing population increase…”? This indicates an increase in population growth rates over time. However, if I am not mistaken, your scenario simulated the current population dynamics in the absence of any fishing mortality. In that case, your scenario would show population growth or an increase in population size. If that is correct, please reword the sentence to clarify.
Tables 6 & 7 – I suggest you remove the cases. Throughout the manuscript you keep your two reproductive cycles separated. The scenarios within each reproductive cycle are the same. Referring to the different scenarios in the tables and graphs that follow will make the comparisons between the reproductive cycles much easier to follow. If you had created single-panel graphs, this discrimination would be necessary, but still difficult to follow. On my first read, I missed the cases in this table and got confused when I came to the results. Make sure your table format is consistent. In table 7 some scenario rows have additional horizontal lines.
Line 336 – What are the stage-based models? Are those the models simulating no fishing mortality? Or are all models stage-based models? It appears you are referring to the no-fishing models here. If so, please clarify. If those two are in fact the only stage-based models, what are the other models then? This needs a bit more clarification.
Line 343 – Remove “Furthermore,…”
Line 345 – 347 – Consider rewording this sentence a bit to improve the flow. For example: “Our results further indicate that under scenario E (0.2 Ms; λ of >1) and the given harvest conditions (F = 20% of M), a population of mako sharks would be stable.”
Line 349 – I do not quite understand why the 3-year reproductive cycle model underestimated the population growth rate compared to the 2-year reproductive cycle. With females spending one year resting instead of moving back to being pregnant straight away following the birth of their pups, wouldn’t you expect a lower population growth rate? If so, why would you underestimate it? I cannot see where your results would support this claim.
Line 361 – I suggest you use a) and b) to refer to the different figure panels to make it easier to follow.
Line 365 – Are the harvest strategy scenarios you are referring to here the previously defined scenarios B-E? If so, I suggest you refer to them the same way as before to avoid confusion. If these are different scenarios, you must make sure you properly define them. Why the double negative? Please correct this. There are either no significant differences, or there are significant differences. Which one did you intend to state?
Figures 4 & 5 – For both figures, I suggest you use a) and b) to label the different panels, making it a bit easier to follow the in-text references to the graphs. Also, I suggest you increase the in-graph font size. It is very difficult to identify the different life history stages correctly. Furthermore, I suggest you use the same scales for the y-axis. This will create empty space in some of the panels but will allow for a much easier comparison between the different scenarios and the two reproductive cycles. This would make the differences described in the text much more apparent and easier to spot.
Line 374 – What is the stage model? I mentioned this earlier; you need to clearly define your models and make sure that it is clear which model you are referring to.
Line 391 – How do you know the 2-year reproductive cycle model overestimated the management measurements? This statement can only be made if you know for certain that the values you received from the 3-year reproductive cycle model are correct. Since both models deal with the same level of uncertainty, I thought the comparison between both models was a relative one. If I am mistaken, please correct me. In this case, you should add an explanation of how to reach this conclusion.
Figure 6 – Please see my comments on tables 6 & 7 regarding the cases. Also, I suggest again you use a) and b) to distinguish between the two figure panels. In any case, the panels would be left and right, not top and bottom.
Line 400 – 401 –Please see my previous comments on the use of cases instead of scenarios.
Line 401 – 403 – This sentence is very difficult to follow. Again, using the different cases instead of simply using the scenarios makes it even harder to understand. From what I understand, you are trying to say that within scenario E only female sharks in the 3-year reproductive cycle have a population growth rate below 1.
Line 403 – What do you mean by “…under F…”? I assume you are referring to female sharks. If so, please state this clearly.
Figure 7 – Again, I suggest a) and b). In any case, as before, left and right not top and bottom.
Line 411 – Refer to the respective scenarios instead of cases. You separated the reproductive cycles. There is no need for further separation.
Line 415 – 416 – What do you mean by “…, particularly when λ was extremely lower than 1.”? Extremely lower is not a scientific expression, so I suggest you reword that part of the sentence. Also, isn’t λ the population growth rate? So a low value for female λ would be the result of certain protection measurements, not the cause.
Discussion
Line 422 – 423 – This is a very strong statement. Given the amount of uncertainty and the very limited data available I would refer to your results as an estimate of Ms.
Line 430 – 431 – I would say “…due to the lack of mako shark capture and effort data.”
Line 432 – “…than female sharks…”
Line 436 – “…estimated to be 0.093-0.200 and 0.077-0.242 respectively according to…”
Line 446 – 459 – Whilst I appreciate the honesty, I suggest you move this part to the end of the discussion. You just highlighted the limitations of your study and almost made it sound as if your results aren’t at all reliable. Make sure you acknowledge the limitations and make suggestions for future studies to verify your results, but do not weaken them. Your study and your results are fantastic and of great value. Do not make the reader doubt that. Moving on from this paragraph it is difficult to re-engage the reader and make him value the interpretations of your results since you just told her/him that your results are just preliminary and subject to a large amount of variability and other limiting factors.
Line 480 – 482 – True, a direct application of your results to mako shark populations elsewhere might be difficult or even impossible. However, comparing certain traits between populations, such as sex and size segregation could reveal similarities and differences between populations, which may allow for a relatively quick adjustment of your model.
Line 491 – “…higher compared to those…”
Line 491 – 493 – “However, Yokoi et al. reported demographic parameters higher than the ones found in this study because of age at maturity and longevity differences across the studies oceans.” – Have those parameters been verified in the field for both studies, or are those estimates? Also, which study used the higher age at maturity and the higher longevity values?
Line 525 – The first sentence feels a bit lost. Consider removing it.
Line 538 – I would make this entire section the last section before the conclusion. You do not want to discuss the limitations of your study before discussing the ecological and management implications of your results. I further suggest you shorten this section a bit. You provide a lot of detail on how you compensated for different uncertainties, making it sound more like a part of the methods than the discussion. If you feel it is important to incorporate the type of models you used, keep it to the methods section. In the discussion focus on how biases may affect the results, not only from a statistical point of view, but from an ecological and management point of view.
Line 546 – 547 – “Our analysis was based only on the relevant biological parameters derived for mako sharks in the southeastern Pacific Ocean (Cerna and Licandeo).”
Line 559 – 560 – Is it a surprise that a 2-year reproductive cycle model yields more optimistic results compared to a 3-year reproductive cycle model? What does this mean for management? Are there any reports on a 2-year reproductive cycle in mako shark populations? If not, is a model based on a 2-year reproductive cycle feasible? Why not just use the 3-year reproductive cycle model?
Line 564 – “…employed in Duffy and Francis, additional studies…”
Line 584 – 585 – “Obtaining biological data, including information on the longevity of a species is difficult, and the lack of these data has hindered demographic analyses of shark populations.”
Line 592 – 593 – “…may reflect the actual conditions more closely.”
Line 594 – “…must be verified in the future.”
Line 598 – “Therefore, we did not use…” – why do you think the observed longevity was lower compared to other oceans? Does it necessarily mean that the observed longevity in the South Pacific Ocean is an underestimate? Could there be other reasons? Would it be worth incorporating the observed longevity in your models to see how it may affect the results?
Line 621 – “…Southwest Pacific Ocean…” – keep it consistent.
Line 639 – 640 – Doesn’t an unstable stock indicate overfishing?
Line 644 – Remove “…,that is,…”; “…sustain a low fishing mortality…”
Line 645 – 646 – Do not mention that the results are preliminary. You have mentioned it before.
Line 646 – 648 – You have mentioned the difficulties related to the assessment before. Make sure you do not repeat yourself too often.
Line 650 – 654 – This sounds like a part of your methods section. Do not just repeat what you wrote there. If there are parts that were not in the methods, but were added here, move them to the methods.
Line 654 – 656 – This is just a very short repeat of your results. But what does this mean?
Line 656 – “…of mako shark decreased.”
Line 672 – Was it observed or predicted to diminish?
Line 679 – Remove “However, this study has several limitations.”
Conclusion
Line 674 – This conclusion needs a bit of work. The only conclusive statement based on your study is the second half of the last sentence. What else did you learn from your study? What are the take home messages? Do not focus on the limitations of your study here, but on the strong results it yielded. How can your modelling approach be used in the future? How can it be adapted to different regions and possibly different species?
Author Response
Responses Reviewer 3's Comments
Major Comments:
Figures
Overall, I suggest you increase the in-graph font size to improve readability. The text on some graphs is impossible to read. I provide further details in the comments below.
Response: We acknowledge this comment and these points have been taken care in revised section. All graphs in this MS have been revised as suggested.
Discussion
I feel this is the part that needs the most work. You tend to repeatedly list the limitations of your study and the difficulties in assessing the stock of mako sharks in the South Pacific Ocean. I recommend you move those issues to a single section of the discussion. Also, avoid being too negative about the results of your study. Yes, your study has some limitations, every study does. But be sure not to make it sound as if the results are completely unreliable and that what I just read is nothing but a preliminary pilot study. You may lose your reader’s attention. This is also the reason why I suggest you move the section about the limitations of your study to the end of the discussion. Make sure this section isn’t too long. You could easily shorten it a bit by removing the parts on the types of models you used and the data you used. This should all be in the methods and there is no need repeating it all in the discussion. If part of what you wrote in the discussion is missing in the methods section, make sure you add it there. Instead of explaining what you did again, try to focus on the implications these biases and limitations may have and how we could reduce these uncertainties in the future. How did other researchers deal with the limitations of their studies?
Response: We acknowledge this comment and the sentences regarding the limitations of this study have been moved to discussion section in current version.
Overall, I think you need to interpret your results a bit more from an ecological perspective. You mentioned mako shark stocks that are almost extinct. Have there been any models predicting the population decline? How accurate were those estimates? How could your model help predict population dynamics of mako sharks in the future? Could your model be adjusted to fit mako shark populations elsewhere or even different species? Your model predicts that protecting sub-adults will have a greater impact on the population growth rate than protecting adults. What does this mean for management? Are mako sharks mostly bycatch or is there a fishery targeting this species?
Response: We acknowledge this comment and these points have been included in revised Discussion section.
Minor Comments:
Simple Summary
Line 10 – Statistical data of what kind? Please be more specific.
Response: This sentence has been changed to “Statistical of baseline and species-specific data are largely unavailable for several elasmobranchs (e.g., sharks, skates, and rays).”
Line 16 – “…fishery data.”
Response: Revised as suggested.
Abstract
Line 31 – 33 – This sentence needs to be reworded. Firstly, you use the term demonstrate twice, so I would suggest you use a synonym for one of them. Secondly, to make the conclusive statement a bit easier to follow, I suggest you state “…reproductive cycle compared to the 3-year reproductive cycle.”.
Response: This sentence has been reworded as suggested. See as below:
“Under unfished conditions, the simulations demonstrated that the mako sharks evidencedemon-strated a higher finite population growth rate in the 2-year reproductive cycle compared tothan in the 3-year reproductive cycle.”
Introduction
Line 50 – “…(IATTC) has yet to report its shark stock assessment…”
Response: Revised as suggested.
Line 54 – consider changing it to “…fast-swimming species, and a vital component of…”
Response: Revised as suggested. See as below:
“The mako shark (Isurus oxyrinchus Rafinesque, 1810, from the family Lamnidae) is a fast-swimming, and a vital component of the pelagic shark community and is widely distributed in tropical and temperate waters with a temperature of >16°C [4,5].”
Line 57 – 58 – I am missing the conclusion of this sentence. You state that I. oxyrinchus has several life history characteristics, then list them. But doesn’t every species have several life history characteristics? In the next sentence, you are pointing out the consequences of the mako shark’s long life, late maturity, late fecundity, and extended reproductive cycle. I suggest you either combine the two sentences into one, which would probably make it quite lengthy, or you change the structure of this sentence, so it simply introduces the characteristics. For example: “This species is known for its long life, low fecundity, late maturity, and extended reproductive cycle (2-3 years).”
Response: This sentence has combined the two sentences into one, as suggested.
Line 59 – You started the previous sentence with “This species…”. Using the term again feels repetitive. Maybe use the species name instead. You could also refer to it as “it” since this statement follows up on the previous sentence.
Response: This sentence has been replaced to “it” as suggested.
Line 66 – 69 – Do you have any reference for this statement? Any previous stock assessments or reports on the uncertainty in fisheries and biological information?
Response: Since this statement is based on previously studies in the Atlantic, the Northwestern Pacific Ocean, and the Indian Ocean. However, apart from two recent preliminary shortfin mako shark stock assessment reports in Southwest Pacific (Large et al. 2022a,b), its population status remains poorly known. In addition, we included several references for biological information in this MS as suggested. See as below:
“However, apart from two recent preliminary mako shark stock assessment reports in the Southwest Pacific [128,129], its population status remains poorly known, as well as in terms of biological information, only three investigations have been conducted to date [50,51,52].”
Large, K.; Neubauer, P.; Brouwer, S., and Kai, M. Stock assessment of Southwest Pacific Shortfin Mako shark. 2022 (WCPFC-SC18-2022/SA-IP-13). [128]
Large, K.; Neubauer, P.; Brouwer, S. Input data for the 2022 South Pacific Shortfin Mako Shark stock assessment. 2022 (WCPFC-SC18-2022/SA-WP-02). [129]
Line 69 – 70 – Delete this sentence. You are still providing a background for your study. You have introduced the knowledge gap. Now you are providing the methods that may be used to close that gap with examples. Your study, including your motivation to do it, should come last.
Response: This sentence has been deleted as suggested.
Line 70 – Remove “However,…”. Start the sentence with “Obtaining the long term…”
Response: This sentence has been removed “However,…” as suggested.
Line 83 – 88 – You should provide some references for these three sentences.
Response: This MS has been provided some references as suggested. See as below:
- Simpfendorfer, C. A. Demographic models: life tables, matrix models and rebound potential. In: Musick, J.A., Bonfil, R. (Eds.), Elas-mobranch Fisheries Management Techniques. APEC Secretariat, Singapore, 2004; pp. 187–204.
- Simpfendorfer, C. A. Demographic models: life tables, matrix models and rebound potential. Management Techniques for Elasmo-branch Fisheries, Eds., by J. A. Musick, and R. Bonfil. Fisheries Technical Paper (474), FAO, Rome, Italy, 2005; pp. 143–144.
- Smart, J. J.; Chin, A.; Tobin, A.; White, W.; Kumasi, B.; Simpfendorger, C. A. Stochastic demographic analyses of the silvertip shark, Carcharhinus albimarginatus, and the common blacktip shark, Carcharhinus limbatus, from the Indo-Pacific. Fish. Res. 2017, 191, 95–107.
Line 95 – Please provide the scientific names for boarfish, electric knifefish, and European sea bass.
Response: Provided the scientific names as suggested in the current version. See as below:
boarfish (Capros aper), electric knifefish (Gymnorhamphichthys rondoni) and European sea bass (Dicentrarchus labrax L.)
Materials and Methods
Figure 1 – The text within the graph is too small to read. Please increase the font size of the scale a bit.
Response: Figure 1 has been changed as suggested.
Table 2 – If the gestation period is two years, why is the expected duration of the pregnant adult stage only one year? Also, if parturient, by definition, refers to the stage of giving birth, an expected duration of one year appears to be very long. My question, in this case, would be if this duration was chosen due to the nature of the model you used. Furthermore, I wonder if you could include parturient females in the pregnant stage.
Response: According to Table 2, the adults (both 2-reproductive cycles) included: pregnant adults, Parturient adults, and Resting adults (only related to the 3-year reproductive cycle). In this study, "parturient adults" defined females who were about to create new neonates (the procedure could be extended by a year) without referring to the stage of giving birth. We also assumed to include parturient females in the pregnant stage in this study.
Line 153 – What is an “ungenerous technique”?
Line 154 – What do you mean by “allowed injuries”?
Response: In this MS, the term "ungenerous approach" meant that the demographic study could have used age-structured or size-structured population dynamic models. However, age-specific data for only a few aquatic species at risk are available, and direct estimates of age-specific vital rates for the mako shark are limited. As a result, the employment of a stage-structured model in this study was encouraged. This is why, in the lack of evidence, appeal for an "ungenerous technique" to estimate "allowable injuries" or "allowable losses".
Line 174 – “…in the 2-year reproductive cycle…”
Response: Revised as suggested. See as below
“Here, we established two-sex models in which the abundance of both male and female sharks influences fecundity [19,41,45] in the 2-year reproductive cycle and the 3-year reproductive cycle with a 1-year resting period;…”
Equation 6 – The numbering for this equation is missing. Also, the subscript and superscript text is very difficult and partly impossible to read. Consider increasing the font size slightly for the equations to increase readability.
Response: This point has been taken care in this version.
Line 227 – 228 – How is the natural mortality rate closely related to exploitation rates and management variables?
Response: Natural mortality is closely related to exploitation rates and management variables, that is, natural mortality may decrease with increased stock exploitation due to lower density and less competition, and natural mortality may vary significantly from year to year depending on changes in predation, food availability, diseases, and other biological and environmental factors. As a result, it is common practice in stock assessment to assume constant M for fishable age classes, as this parameter is notoriously difficult to evaluate directly.
Line 288 – “…was set to equal 80%...” – change this throughout the list of scenarios.
Response: Revised as suggested. However, we removed "equals" from each bullet point as they were unnecessary.
Results
Line 314 – “…values than female sharks…”
Response: Removed “did” as suggested.
Figure 3 – Please provide more information in the figure caption. What does each orange dot represent? What do the vertical blue lines show? Given there is only one dashed red line for each sex, does it represent the overall mean natural mortality value across all 14 estimators? Also, please increase the font size of the in-graph text a bit to improve the readability. I also suggest you change the colour of the 14 estimators. Black text on a dark background makes it unreadable now.
Response: Figure 3 has been changed as suggested. In addition, there is a dashed red line for each sex, it represents the overall mean natural mortality value across all 14 estimators.
Line 329 – What do you mean by “…an increasing population increase…”? This indicates an increase in population growth rates over time. However, if I am not mistaken, your scenario simulated the current population dynamics in the absence of any fishing mortality. In that case, your scenario would show population growth or an increase in population size. If that is correct, please reword the sentence to clarify.
Response: This has been changed as suggested. See as below:
“According to the most optimistic input parameters, the deterministic model of unfished conditions indicated an increasing population with mean and 95% CI for our 2- and 3-year reproductive cycle models, with sex-specificity evident in all scenarios (Tables S2 and S3).”
Tables 6 & 7 – I suggest you remove the cases. Throughout the manuscript you keep your two reproductive cycles separated. The scenarios within each reproductive cycle are the same. Referring to the different scenarios in the tables and graphs that follow will make the comparisons between the reproductive cycles much easier to follow. If you had created single-panel graphs, this discrimination would be necessary, but still difficult to follow. On my first read, I missed the cases in this table and got confused when I came to the results. Make sure your table format is consistent. In table 7 some scenario rows have additional horizontal lines.
Response: We agree with the reviewer's comment. These words has been removed from Tables 6 and 7. The table format has been checked and revised in this version.
Line 336 – What are the stage-based models? Are those the models simulating no fishing mortality? Or are all models stage-based models? It appears you are referring to the no-fishing models here. If so, please clarify. If those two are in fact the only stage-based models, what are the other models then? This needs a bit more clarification.
Response: Demographic matrix population models such as age-structured (also known as Leslie Matrix) and stage-structured models are commonly used in the assessment of shark populations. The choice between age- and stage-structured models is basically depending on personal preference. Both approaches will provide similar results if the same life history parameters are used.
In this study, the demographic stage-structured model was adopted and two conditions including with and without fishing mortality were examined.
Line 343 – Remove “Furthermore,…”
Response: This sentence has been removed as suggested.
Line 345 – 347 – Consider rewording this sentence a bit to improve the flow. For example: “Our results further indicate that under scenario E (0.2 Ms; λ of >1) and the given harvest conditions (F = 20% of M), a population of mako sharks would be stable.”
Response: Revised as suggested.
Line 349 – I do not quite understand why the 3-year reproductive cycle model underestimated the population growth rate compared to the 2-year reproductive cycle. With females spending one year resting instead of moving back to being pregnant straight away following the birth of their pups, wouldn’t you expect a lower population growth rate? If so, why would you underestimate it? I cannot see where your results would support this claim.
Response: We agreed with the reviewer on this issue. In this MS, this sentence has been replaced with "estimated a lower" since the 3-year reproductive cycle model has not been directly tested, therefore "underestimated" is inappropriate.
Line 361 – I suggest you use a) and b) to refer to the different figure panels to make it easier to follow.
Response: We agreed with the reviewer's comment. This MS has been revised as suggested.
Line 365 – Are the harvest strategy scenarios you are referring to here the previously defined scenarios B-E? If so, I suggest you refer to them the same way as before to avoid confusion. If these are different scenarios, you must make sure you properly define them. Why the double negative? Please correct this. There are either no significant differences, or there are significant differences. Which one did you intend to state?
Response: To avoid confusing the readers, this sentence “In the 3-year reproductive cycle model, the RVs in subadult male sharks exceeded those in adult male sharks in harvest strategy scenarios when F = 0.8 and 0.6, without nonsignificant differences (Figure 6, bottom).” has been removed in this version.
Figures 4 & 5 – For both figures, I suggest you use a) and b) to label the different panels, making it a bit easier to follow the in-text references to the graphs. Also, I suggest you increase the in-graph font size. It is very difficult to identify the different life history stages correctly. Furthermore, I suggest you use the same scales for the y-axis. This will create empty space in some of the panels but will allow for a much easier comparison between the different scenarios and the two reproductive cycles. This would make the differences described in the text much more apparent and easier to spot.
Response: Figures 4 and 5 have been revised as suggested.
Line 374 – What is the stage model? I mentioned this earlier; you need to clearly define your models and make sure that it is clear which model you are referring to.
Response: Revised as suggested. Please also see our response to Line 336.
Line 391 – How do you know the 2-year reproductive cycle model overestimated the management measurements? This statement can only be made if you know for certain that the values you received from the 3-year reproductive cycle model are correct. Since both models deal with the same level of uncertainty, I thought the comparison between both models was a relative one. If I am mistaken, please correct me. In this case, you should add an explanation of how to reach this conclusion.
Response: We agreed with the reviewer comments. This sentence has been modified as "estimated a higher" since the 3-year reproductive cycle model has not been directly tested, and hence "overestimated" is incorrect.
Figure 6 – Please see my comments on tables 6 & 7 regarding the cases. Also, I suggest again you use a) and b) to distinguish between the two figure panels. In any case, the panels would be left and right, not top and bottom.
Response: Figure 6 has been changed as suggested.
Line 400 – 401 –Please see my previous comments on the use of cases instead of scenarios.
Response: Revised as suggested.
Line 401 – 403 – This sentence is very difficult to follow. Again, using the different cases instead of simply using the scenarios makes it even harder to understand. From what I understand, you are trying to say that within scenario E only female sharks in the 3-year reproductive cycle have a population growth rate below 1.
Response: Probably this sentence was not well written and the results mislead the readers. To avoid the confusion, this sentence has been revised as “However, λ was the lowest in female sharks (0.991 year−1 on Scenario 5 in the 3-year reproductive cycle model), which was lower than the threshold λ of 1.”
Line 403 – What do you mean by “…under F…”? I assume you are referring to female sharks. If so, please state this clearly.
Response: “…under F…” mean “under the F put into these models”. This sentence has been revised as “Notably, under the F put into these models (Scenarios 2 and 3) in both the reproductive cycle models, all λ values were <1, particularly for female sharks in Scenarios 2–4.”
Figure 7 – Again, I suggest a) and b). In any case, as before, left and right not top and bottom.
Response: Figure 7 has been changed as suggested.
Line 411 – Refer to the respective scenarios instead of cases. You separated the reproductive cycles. There is no need for further separation.
Response: We agreed with the reviewer comments. This has been changed as suggested.
Line 415 – 416 – What do you mean by “…, particularly when λ was extremely lower than 1.”? Extremely lower is not a scientific expression, so I suggest you reword that part of the sentence. Also, isn’t λ the population growth rate? So a low value for female λ would be the result of certain protection measurements, not the cause.
Response: We agreed with the reviewer comments. This sentence has been revised as “…particularly when λ was relatively lower than 1.
Discussion
Line 422 – 423 – This is a very strong statement. Given the amount of uncertainty and the very limited data available I would refer to your results as an estimate of Ms.
Response: This has been changed as suggested. The completed statement will be worded as follows: “In this study, despite limited data availability, we estimated the Ms of mako sharks in the South Pacific Ocean by using indirect (empirical) approaches.”
Line 430 – 431 – I would say “…due to the lack of mako shark capture and effort data.”
Response: Revised as suggested. See as below:
“Although Brodziak et al. [62] and Hoenig et al. [85] recommended comparing indirect es-timates of M with field-derived measurements of M, direct estimates of M are unavailable due to the lack of mako shark capture and effort data.”
Line 432 – “…than female sharks…”
Response: This MS has been removed “did” before “female” as suggested.
Line 436 – “…estimated to be 0.093-0.200 and 0.077-0.242 respectively according to…”
Response: Revised as suggested. As shown below, the completed statement will be written as: “…; moreover, the Ms of male and female mako sharks in the Northwest Pacific Ocean was estimated to be 0.093–0.200 and 0.077–0.242 respectively according to Tsai et al. [18] and 0.119–0.141 and 0.091–0.124 according to Tsai et al. [19].”
Line 446 – 459 – Whilst I appreciate the honesty, I suggest you move this part to the end of the discussion. You just highlighted the limitations of your study and almost made it sound as if your results aren’t at all reliable. Make sure you acknowledge the limitations and make suggestions for future studies to verify your results, but do not weaken them. Your study and your results are fantastic and of great value. Do not make the reader doubt that. Moving on from this paragraph it is difficult to re-engage the reader and make him value the interpretations of your results since you just told her/him that your results are just preliminary and subject to a large amount of variability and other limiting factors.
Response: We agreed with the reviewer’s comment. This part has been moved to the end of the discussion section as suggested.
Line 480 – 482 – True, a direct application of your results to mako shark populations elsewhere might be difficult or even impossible. However, comparing certain traits between populations, such as sex and size segregation could reveal similarities and differences between populations, which may allow for a relatively quick adjustment of your model.
Response: We acknowledge this helpful suggestion and this will be our future research important reference.
Line 491 – “…higher compared to those…”
Response: Revised as suggested.
Line 491 – 493 – “However, Yokoi et al. reported demographic parameters higher than the ones found in this study because of age at maturity and longevity differences across the studies oceans.”
Response: The reviewer is correct. This difference was due to different life history parameters for shortfin mako sharks used in the studies.
Line 525 – The first sentence feels a bit lost. Consider removing it.
Response: This sentence has been removed in this version.
Line 538 – I would make this entire section the last section before the conclusion. You do not want to discuss the limitations of your study before discussing the ecological and management implications of your results. I further suggest you shorten this section a bit. You provide a lot of detail on how you compensated for different uncertainties, making it sound more like a part of the methods than the discussion. If you feel it is important to incorporate the type of models you used, keep it to the methods section. In the discussion focus on how biases may affect the results, not only from a statistical point of view, but from an ecological and management point of view.
Response: We agreed with the reviewer’s comments. This paragraph has been relocated and shortened.
Line 546 – 547 – “Our analysis was based only on the relevant biological parameters derived for mako sharks in the southeastern Pacific Ocean (Cerna and Licandeo).”
Response: Revised as suggested.
Line 559 – 560 – Is it a surprise that a 2-year reproductive cycle model yields more optimistic results compared to a 3-year reproductive cycle model? What does this mean for management? Are there any reports on a 2-year reproductive cycle in mako shark populations? If not, is a model based on a 2-year reproductive cycle feasible? Why not just use the 3-year reproductive cycle model?
Response: The estimated population growth rate was higher for a 2-year reproductive cycle than for a 3-year reproductive cycle. The result is reasonable because the annual stage-specific per-capita fecundity (fi) is higher for the 2-year model than the 3-year model. Mollet et al. (2000) recommended a 3-year reproductive cycle with a 2-year gestation period and a 1-year resting period, although a 2-year cycle could not be ruled out. Consequently, both a 2-year and a 3-year reproductive cycle were examined in the analyses.
Mollet, H. F., Cliff, G., Pratt, H. L., Jr, and Stevens, J. D. 2000. Reproductive biology of the female shortfin mako, Isurus oxyrinchus Rafinesque, 1810, with comments on the embryonic development of lamnoids. Fishery Bulletin, 98: 299–318.
Line 564 – “…employed in Duffy and Francis, additional studies…”
Response: Revised as suggested.
Line 584 – 585 – “Obtaining biological data, including information on the longevity of a species is difficult, and the lack of these data has hindered demographic analyses of shark populations.”
Response: Revised as suggested.
Line 592 – 593 – “…may reflect the actual conditions more closely.”
Response: Revised as suggested.
Line 594 – “…must be verified in the future.”
Response: Revised as suggested.
Line 598 – “Therefore, we did not use…” – why do you think the observed longevity was lower compared to other oceans? Does it necessarily mean that the observed longevity in the South Pacific Ocean is an underestimate? Could there be other reasons? Would it be worth incorporating the observed longevity in your models to see how it may affect the results?
Response: In this study, the range of male longevity (22–31 yrs) was assumed, which includes the observed Tmax (22 and 29) (see table below). However, the range of female longevity (36–45 yrs) was assumed, which does not include the observed Tmax (22 and 28) (see table below).
The female growth parameter (K) was lower than the male which means female should have higher longevity than male shortfin mako shark. This characteristic can also be found in the North Pacific Ocean (Change and Liu, 2009; Tsai et al. 2014, 2015). Therefore, the observed longevity (22 or 28) for female shortfin mako shark is believed to be underestimated in the South Pacific Ocean. Alternatively, the female longevity was estimated based on empirical equations (eq 12).
Chang, J. H., and Liu, K. M. 2009. Stock assessment of the shortfin mako shark (Isurus oxyrinchus) in the Northwest Pacific Ocean using perrecruit and virtual population analyses. Fisheries Research, 98: 92–101.
Tsai, W. P., K. M. Liu, A. E. Punt, and C. L. Sun. 2015. Assessing the potential biases of ignoring sexual dimorphism and mating mechanism in using a single-sex demographic model: the shortfin mako shark as a case study. ICES Journal of Marine Science, 72(3): 793–803.
Tsai, W. P., C. L. Sun., A. E. Punt, and K. M. Liu. 2014. Demographic analysis of the shortfin mako shark, Isurus oxyrinchus, in the Northwestern Pacific using a two-sex stage-based matrix model. ICES Journal of Marine Science, 71(7): 1604–1618.
Sources of uncertainty |
Male |
Female |
Assumed distribution |
Fecundity |
Triangle (8, 12.5, 18) |
Triangle (8, 12.5, 18) |
Triangular distribution |
Natural mortality |
ln (mean, SD)* |
ln (mean, SD)* |
Lognormal |
Age at maturity |
7–10 years |
19–21 years |
Uniform |
Longevity |
22–31 years |
36–45 years |
Uniform |
Line 621 – “…Southwest Pacific Ocean…” – keep it consistent.
Response: Revised as suggested.
Line 639 – 640 – Doesn’t an unstable stock indicate overfishing?
Response: The reviewer is correct. The unstable stock status was due to population decline (overfishing).
Line 644 – Remove “…,that is,…”; “…sustain a low fishing mortality…”
Response: Revised as suggested. See as below:
“Our results revealed mako shark populations, would only be able to sustain a relatively low fishing mortality (20% of natural mortality) in the study area.”
Line 645 – 646 – Do not mention that the results are preliminary. You have mentioned it before.
Response: This has been revised as suggested.
Line 646 – 648 – You have mentioned the difficulties related to the assessment before. Make sure you do not repeat yourself too often.
Response: This has been taken care of in this version.
Line 650 – 654 – This sounds like a part of your methods section. Do not just repeat what you wrote there. If there are parts that were not in the methods, but were added here, move them to the methods.
Response: We agreed with the reviewer’s comment. This has been taken care of in this version.
Line 654 – 656 – This is just a very short repeat of your results. But what does this mean?
Response: We agreed with the reviewer’s comment. An additional sentence has been added in this version to explain more specifically (See sentence below)
“This result also showed that any added anthropogenic source of mortality will likely result in its population decline.”
Line 656 – “…of mako shark decreased.”
Response: Revised as suggested.
Line 672 – Was it observed or predicted to diminish?
Response: The wording "predicted" has been replaced.
Line 679 – Remove “However, this study has several limitations.”
Response: “However” has been removed as suggested.
Conclusion
Line 674 – This conclusion needs a bit of work. The only conclusive statement based on your study is the second half of the last sentence. What else did you learn from your study? What are the take home messages? Do not focus on the limitations of your study here, but on the strong results it yielded. How can your modelling approach be used in the future? How can it be adapted to different regions and possibly different species?
Response: We agree with reviewer’s comments. The additional sentences have been added in revised Conclusion section (see sentences below)
“Better estimates of natural and fishing mortality are required to understand the impact of commercial fisheries in the South Pacific Ocean's shortfin mako shark population. It is also suggested that our approach can be utilized as an assessment tool for shark species with insufficient catch and effort data.”
